# Similarity-based cooperative equilibrium

**Caspar Oesterheld**[1][*]    **Johannes Treutlein**[2]    **Roger Grosse**[3,4,5]
**Vincent Conitzer**[1,6]    **Jakob Foerster**[7]
[*]`oesterheld@cmu.edu`   [1]FOCAL, Carnegie Mellon University   [2]CHAI, UC Berkeley
[3]Anthropic   [4]Vector Institute   [5]University of Toronto
[6]Institute for Ethics in AI, University of Oxford   [7]FLAIR, University of Oxford

## Abstract

As machine learning agents act more autonomously in the world, they will increasingly interact with each other. Unfortunately, in many social dilemmas like the one-shot Prisoner's Dilemma, standard game theory predicts that ML agents will fail to cooperate with each other. Prior work has shown that one way to enable cooperative outcomes in the one-shot Prisoner's Dilemma is to make the agents mutually transparent to each other, i.e., to allow them to access one another's source code (Rubinstein, 1998; Tennenholtz, 2004) – or weights in the case of ML agents. However, full transparency is often unrealistic, whereas partial transparency is commonplace. Moreover, it is challenging for agents to learn their way to cooperation in the full transparency setting. In this paper, we introduce a more realistic setting in which agents only observe a single number indicating how similar they are to each other. We prove that this allows for the same set of cooperative outcomes as the full transparency setting. We also demonstrate experimentally that cooperation can be learned using simple ML methods.

## 1 Introduction

As AI systems start to autonomously interact with the world, they will also increasingly interact with each other. We already see this in contexts such as trading agents (CFTC & SEC, 2010), but the number of domains where separate AI agents interact with each other in the world is sure to grow; for example, consider autonomous vehicles. In the language of game theory, AI systems will play general-sum games with each other. For example, autonomous vehicles may find themselves in Game-of-Chicken-like dynamics with each other (cf. Fox et al., 2018). In many of these interactions, cooperative or even peaceful outcomes are not a given. For example, standard game theory famously predicts and recommends defecting in the one-shot Prisoner's Dilemma. Even when cooperative equilibria exist, there are typically many equilibria, including uncooperative and asymmetric ones. For instance, in the infinitely repeated Prisoner's Dilemma, mutual cooperation is played in some equilibria, but so is mutual defection, and so is the strategy profile in which one player cooperates 70% of the time while the other cooperates 100% of the time. Moreover, the strategies from different equilibria typically do not cooperate with each other. A recent line of work at the intersection of AI/(multi-agent) ML and game theory aims to increase AI/ML systems' ability to cooperate with each other (Stastny et al., 2021; Dafoe et al., 2020; Conitzer & Oesterheld, 2023).

Prior work has proposed to make AI agents *mutually transparent* to allow for cooperation in equilibrium (McAfee 1984; Howard 1988; Rubinstein 1998, Section 10.4; Tennenholtz 2004; Barasz et al. 2014; Critch 2019; Oesterheld 2019b). Roughly, this literature considers for any given 2-player normal-form game $\Gamma$ the following *program meta game*: Both players submit a computer program, e.g., some neural net, to choose actions in $\Gamma$ on their behalf. The computer program then receives as input the computer program submitted by the other player. The aforecited works have shown that the program meta game has cooperative equilibria in the Prisoner's Dilemma.

Unfortunately, there are multiple obstacles to cooperation based on full mutual transparency. 1) Settings of *full* transparency are rare in the real world. 2) Games played with full transparency in

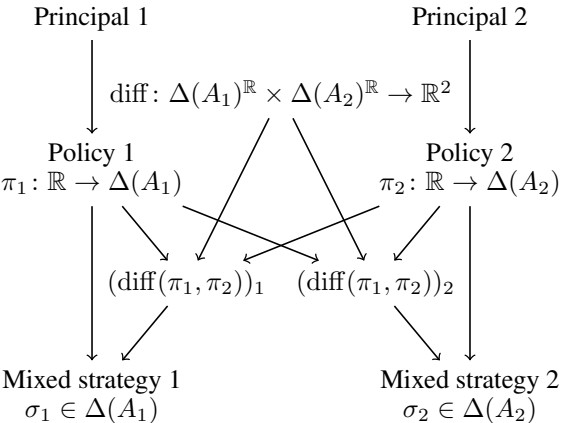

Figure 1: A graphical representation of diff meta games (Definition 1). Nodes with two incoming nodes are determined by applying one of the parent nodes to the other.

general have many equilibria, including ones that are much worse for some or all players than the Nash equilibria of the underlying game (see the folk theorems given by Rubinstein 1998, Section 10.4, and Tennenholtz 2004). In particular, full mutual transparency can make the problem of equilibrium selection very difficult. 3) The full transparency setting poses challenges to modern ML methods. In particular, it requires at least one of the models to receive as input a model that has at least as many parameters as itself. Meanwhile, most modern successes of ML use models that are orders of magnitudes larger than the input. Consequently, we are not aware of successful projects on learning general-purpose models such as neural nets in the full transparency setting.

**Contributions.** In this paper we introduce a novel variant of program meta games called *difference (diff) meta games* that enables cooperation in equilibrium while also addressing obstacles 1–3. As in the program meta game, we imagine that two players each submit a program or *policy* to instruct an *agent* to play a given game, such as the Prisoner's Dilemma. The main idea is that before choosing an action, the agents receive credible information about how *similar* the two players' policies are to each w.r.t. how they make the present decision. In the real world, we might imagine that this information is provided by a mediator (cf. Monderer & Tennenholtz, 2009; Ivanov et al., 2023; Christoffersen et al., 2023) who wants to enable cooperation. We may also imagine that this signal is obtained more organically. For example, we might imagine that the agents can see that their policies were generated using the same code base. We formally introduce this setup in Section 3. Because it requires a much lower degree of mutual transparency, we find the diff meta game setup more realistic than the full mutual transparency setting. Thus, it addresses Obstacle 1 to cooperation based on full mutual transparency.

Diff meta games can still have cooperative equilibria when the underlying base game does not. Specifically, in Prisoner's Dilemma-like games, there are equilibria in which both players submit policies that cooperate with similar policies and thus with each other. We call this phenomenon *similarity-based cooperation (SBC)*. For example, consider the Prisoner's Dilemma as given in Table 1 for $G = 3$. (We study such examples in more detail in Section 3.) Imagine that the players can only submit *threshold policies* that are parameterized only by a single real-valued threshold $\theta_i$ and cooperate if and only if the perceived difference to the opponent is at most $\theta_i$. As a measure of difference, the policies observe $\mathrm{diff}(\theta_1, \theta_2) = |\theta_1 - \theta_2| + Z$, where $Z$ is sampled independently for each player according to the uniform distribution over $[0, 1]$. For instance, if Player 1 submits a threshold of $1/2$ and Player 2 submits a threshold of $3/4$, then the perceived difference is $1/4 + Z$. Hence, Player 1 cooperates with probability $P(1/4 + Z \leq 1/2) = 1/4$ and Player 2 cooperates with probability $P(1/4 + Z \leq 3/4) = 1/2$. It turns out that $(\theta_1 = 1, \theta_2 = 1)$, which leads to mutual cooperation with probability 1, is a Nash equilibrium of the meta game. Intuitively, the only way for either player to defect more is to lower their threshold. But then $|\theta_1 - \theta_2|$ will increase, which will cause the opponent to defect more (at a rate of $1/2$). This outweighs the benefit of defecting more oneself.

In Section 4, we prove a folk theorem for diff meta games. Roughly speaking, this result shows that observing a diff value is sufficient for enabling all the cooperative outcomes that full mutual

|          |           | Player 2 | |
|          |           | Cooperate | Defect |
| Player 1 | Cooperate | $G, G$ | $0, G+1$ |
|          | Defect    | $G+1, 0$ | $1, 1$ |

Table 1: The Prisoner's Dilemma, parameterized by some number $G > 1$.

transparency enables. Specifically, we show that for every *individually rational* strategy profile $\boldsymbol{\sigma}$ (i.e., every strategy profile that is better for each player than their minimax payoff), there is a function diff such that $\boldsymbol{\sigma}$ is played in an equilibrium of the resulting diff meta game.

Next, we address Obstacle 2 to full mutual transparency – the multiplicity of equilibria. First, note that any given measure of similarity will typically only enable a specific set of equilibria, much smaller than the set of individually rational strategy profiles. For instance, in the above example, all equilibria are symmetric. In general, one would hope that similarity-based cooperation will result in symmetric outcomes in symmetric games. After all, the new equilibria of the diff game are based on submitting similar policies and if two policies play different strategies against each other, they cannot be similar. In Section 5, we substantiate this intuition. Specifically, we prove, roughly speaking, that in symmetric, additively decomposable games, the Pareto-optimal equilibrium of the meta game is unique and gives both players the same utility, if the measure of difference between the agents satisfies a few intuitive requirements (Section 5). For example, in the Prisoner's Dilemma, the unique Pareto-optimal equilibrium of the meta game must be one in which both players cooperate with the same probability.

Finally we show that diff meta games address Obstacle 3: we demonstrate that in games with higher-dimensional action spaces, we can find cooperative equilibria of diff meta games with ML methods. In Section 6.4, we show that, if we initialize the two policies randomly and then let each of them learn to be a best response to the other, they generally converge to the Defect-Defect equilibrium. This is expected based on results in similar contexts, such as in the Iterated Prisoner's Dilemma. However, in Section 6.1, we introduce a novel, general pretraining method that trains policies to cooperate against copies and defect (i.e., best respond) against randomly generated policies. Our experiments show that policies pretrained in this way find partially cooperative equilibria of the diff game when trained against each other via alternating best response training.

We discuss how the present paper relates to prior work in Section 7. We conclude in Section 8 with some ideas for further work.

## 2 Background

**Elementary game theory definitions.** We assume familiarity with game theory. For an introduction, see Osborne (2004). A *(two-player, normal-form) game* $\Gamma = (A_1, A_2, \mathbf{u})$ consists of sets of actions or pure strategies $A_1$ and $A_2$ for the two players and a utility function $\mathbf{u} \colon A_1 \times A_2 \to \mathbb{R}^2$. Table 1 gives the Prisoner's Dilemma as a classic example of a game. A mixed strategy for Player $i$ is a distribution over $A_i$. We denote the set of such distributions by $\Delta(A_i)$. We can extend $\mathbf{u}$ to mixed strategies by taking expectations, i.e., $\mathbf{u}(\sigma_1, \sigma_2) \coloneqq \sum_{a_1 \in A_1, a_2 \in A_2} \sigma_1(a_1)\sigma_2(a_2)\mathbf{u}(a_1, a_2)$. For any player $i$, we use $-i$ to denote the other player. We call $\sigma_i$ a *best response* to a strategy $\sigma_{-i} \in \Delta(A_{-i})$, if $\mathrm{supp}(\sigma_i) \subseteq \arg\max_{a_i \in A_i} u_i(a_i, \sigma_{-i})$, where $\mathrm{supp}$ denotes the support. A strategy profile $\boldsymbol{\sigma} \in \Delta(A_1) \times \Delta(A_2)$ is a vector of strategies, one for each player. We call a strategy profile $(\sigma_1, \sigma_2)$ a *(strict) Nash equilibrium* if $\sigma_1$ is a (unique) best response to $\sigma_2$ and *vice versa*. As first noted by Nash (1950), each game has at least one Nash equilibrium. We say that a strategy profile $\boldsymbol{\sigma}$ is *individually rational* if each player's payoff is at least her minimax payoff, i.e., if $u_i(\boldsymbol{\sigma}) \geq \min_{\sigma_{-i} \in \Delta(A_{-i})} \max_{a_i \in A_i} u_i(a_i, \sigma_{-i})$ for $i = 1, 2$. We say that $\boldsymbol{\sigma}$ is *Pareto-optimal* if there exists no $\boldsymbol{\sigma}'$ s.t. $u_i(\boldsymbol{\sigma}') \geq u_i(\boldsymbol{\sigma})$ for $i = 1, 2$ and $u_i(\boldsymbol{\sigma}') > u_i(\boldsymbol{\sigma})$ for at least one $i$.

**Symmetric games and additively decomposable games.** We say that a game is *(player) symmetric* if $A_1 = A_2$ and for all $a_1, a_2$ for $i = 1, 2$, we have that $u_i(a_1, a_2) = u_{-i}(a_2, a_1)$. The Prisoner's Dilemma in Table 1 is symmetric. We say that a game *additively decomposes into* $(u_{i,j} \colon A_j \to \mathbb{R})_{i,j \in \{1,2\}}$ if $u_i(a_1, a_2) = u_{i,1}(a_1) + u_{i,2}(a_2)$ for all $i = \{1, 2\}$ and all $a_1 \in A_1, a_2 \in A_2$. Intuitively, this means that each action $a_j$ of Player $j$ generates some amount of utility $u_{i,j}(a_j)$ for Player $i$ *independently* of what Player $-j$ plays. For example, the Prisoner's Dilemma in Table 1 is additively decomposable, where $u_{i,i} \colon \mathrm{Cooperate} \mapsto 0, \mathrm{Defect} \mapsto 1$ and $u_{i,-i} \colon \mathrm{Cooperate} \mapsto$

$G$, Defect $\mapsto 0$ for $i = 1, 2$. Intuitively, Cooperate generates $G$ for the opponent and $0$ for oneself, while Defect generates $1$ for oneself and $0$ for the opponent.

**Alternating best response learning.** The orthodox approach to learning in games is to learn to best respond to the opponent, essentially ignoring that the opponent is also a learning agent. In this paper, we specifically consider alternating best response (ABR) learning. In ABR, the players take turns. In each turn, one of the two players updates the parameters $\boldsymbol{\theta}_i$ of her strategy to optimize $u_i(\boldsymbol{\theta}_i, \boldsymbol{\theta}_{-i})$, i.e., updates her model to be a best response to the opponent's current model (Brown cf. 1951; Zhang et al. 2022; Heinrich et al. 2023). Since learning an exact best response is generally intractable, we will specifically consider the use of gradient ascent in each turn to optimize $u_i(\boldsymbol{\theta}_i, \boldsymbol{\theta}_{-i})$ over $\boldsymbol{\theta}_i$. In continuous games if ABR with *exact* (locally) best response updates converges to $(\boldsymbol{\theta}_1, \boldsymbol{\theta}_2)$, then $(\boldsymbol{\theta}_1, \boldsymbol{\theta}_2)$ is a (local) Nash equilibrium. Note, however, that ABR may fail to converge (e.g., in the face of Rock–Paper–Scissors dynamics). Moreover, if the best response updates of $\theta_i$ are only approximated, ABR may converge to non-equilibria (Mazumdar et al., 2020, Proposition 6).

## 3 Diff Meta Games

We now formally introduce diff meta games, the novel setup we consider throughout this paper. Given some base game $\Gamma$, we consider a new *meta game* played by two players whom we will call *principals*. Each principal $i$ submits a *policy*. The two players' policies each observe a real-valued measure of how similar they are to each other. Based on this, the policies then output a (potentially mixed) strategy for the base game. Finally, the utility is realized as per the base game. Below we define this new game formally. This model is illustrated in Figure 1.

**Definition 1.** *Let $\Gamma = (A_1, A_2, \mathbf{u})$ be a game. A (diff-based) policy for Player $i$ for $\Gamma$ is a function $\pi \colon \mathbb{R} \to \Delta(A_i)$ mapping the perceived real-valued difference between the diff-based policies to a mixed strategy of $\Gamma$. For $i = 1, 2$ let $\mathcal{A}_i \subseteq \Delta(A_i)^{\mathbb{R}}$ be a set of difference-based policies for Player $i$. Then a policy difference (diff) function for $(\mathcal{A}_1, \mathcal{A}_2)$ is a stochastic function $\mathrm{diff} \colon \mathcal{A}_1 \times \mathcal{A}_2 \rightsquigarrow \mathbb{R}^2$. For any two policies $\pi_1, \pi_2$ and difference function $\mathrm{diff}$, we say that $(\pi_1, \pi_2)$ plays the strategy profile $\boldsymbol{\sigma} \in \Delta(A_1) \times \Delta(A_2)$ of $\Gamma$ if $\sigma_i = \mathbb{E}\left[\pi_i(\mathrm{diff}_i(\pi_1, \pi_2))\right]$ for $i = 1, 2$. For sets of policies $\mathcal{A}_1, \mathcal{A}_2$ and difference function $\mathrm{diff}$ we then define the diff meta game $(\Gamma, \mathcal{A}_1, \mathcal{A}_2, \mathrm{diff})$ to be the normal-form game $(\mathcal{A}_1, \mathcal{A}_2, V)$, where $V(\pi_1, \pi_2) \coloneqq \mathbb{E}\left[\mathbf{u}((\pi_i(\mathrm{diff}_i(\pi_1, \pi_2)))_{i=1,2})\right]$ for all $\pi_1 \in \mathcal{A}_1, \pi_2 \in \mathcal{A}_2$.*

Note that Definition 1 does not put any restrictions on $\mathrm{diff}$. For example, the above definition allows $(\mathrm{diff}(\pi_i, \pi_{-i}))_i$ to be a real number whose binary representation uniquely specifies $\pi_{-i}$. This paper is dedicated to situations in which $\mathrm{diff}$ specifically represents some intuitive notion of how different the policies are, thus excluding such $\mathrm{diff}$ functions. Unfortunately, there are many different ways in which one could formalize this constraint, especially in asymmetric games. In Section 5 we will impose some restrictions along these lines, including symmetry. Our folk theorem (Theorem 3 in Section 4) will similarly impose constraints on $\mathrm{diff}$ to avoid $\mathrm{diff}$ functions like the above.

The rest of this section will study concrete examples of Definition 1. First, we define a particularly simple type of diff-based policy. Almost all of our theoretical analysis will be based on this class of policies.

**Definition 2.** *Let $\theta \in \mathbb{R} \cup \{-\infty, \infty\}$ and $\sigma_i^{\leq}, \sigma_i^{>} \in \Delta(A_i)$ be strategies for Player $i$ for $i = 1, 2$. Then we define $(\sigma_i^{\leq}, \theta, \sigma_i^{>})$ to be the policy $\pi$ s.t. $\pi(d) = \sigma_i^{\leq}$ if $d \leq \theta$ and $\pi(d) = \sigma_i^{>}$ otherwise. We call policies of this form threshold policies. Let $\bar{\mathcal{A}}_i$ denote the set of such threshold policies.*

Throughout the rest of this section, we analyze the Prisoner's Dilemma as a specific example. We limit attention to threshold agents of the form $(C, \theta, D)$, i.e., policies that cooperate against similar opponents (diff below threshold $\theta$) and defect against dissimilar opponents. This is because such policies can be used to form cooperative equilibria, while policies that always cooperate (say, $(C, 1, C)$) or policies that are more cooperative against *less* similar opponent policies (e.g., $(D, 1, C)$) cannot be used to form cooperative equilibria in the PD with a natural diff function. Policies of the form $(C, \theta, D)$ are uniquely specified by a single real number $\theta$. A natural measure of the similarity between two policies $\theta_1, \theta_2$ is then the absolute difference $|\theta_1 - \theta_2|$. We allow $\mathrm{diff}$ to be noisy, however. We summarize this in the following.

**Example 1.** *Let $\Gamma$ be the Prisoner's Dilemma as per Table 1. Then consider the $(\Gamma, \hat{A}_1, \hat{A}_2, \mathrm{diff})$ meta game where $\hat{A}_i = \{(C, \theta_i, D) \mid \theta_i \in \mathbb{R}\}$ and $\mathrm{diff}_i((C, \theta_1, D), (C, \theta_2, D))) = |\theta_1 - \theta_2| + Z_i$ for $i = 1, 2$ where $Z_i$ is some real-valued random variable.*

The only open parameters of Example 1 are $G$ (the parameter used in our definition of the Prisoner's Dilemma) and the noise distribution. Nevertheless, Example 1 is a rich setting that allows for nontrivial results. We leave a detailed analysis for Appendix B and only give two specific results about equilibria here.

In the first result, we imagine that the noise $Z_i$ is distributed uniformly between 0 and $\epsilon > 0$ and that $G$ is at least 2. Then, roughly, there are two kinds of equilibria. First, there are equilibria in which both players always defect, because their threshold for cooperation is at most 0 (such that they defect with probability 1 even against exact copies). Second, and more interestingly, there are equilibria in which both players submit *the same* threshold strictly between 0 and $\epsilon$. Note that this means that if both players submit a threshold of $\epsilon$, they both cooperate with probability 1.

**Proposition 1.** *Consider Example 1 with $Z_i \sim \mathrm{Uniform}([0, \epsilon])$ i.i.d. for some $\epsilon > 0$ and with $G \geq 2$. Then $((C, \theta_1, D), (C, \theta_2, D))$ is a Nash equilibrium if and only if $\theta_1, \theta_2 \leq 0$ or $0 < \theta_1 = \theta_2 \leq \epsilon$. In case of the latter, the equilibrium is strict if $G > 2$.*

What happens if, instead of the uniform distribution, we let the $Z_i$ be, say, normally distributed? It turns out that for all unimodal distributions (which includes the normal distribution) and $G = 2$, we get an especially simple result: in equilibrium, both players submit the same threshold and that threshold must be left of the mode.

**Proposition 2.** *Consider Example 1 with $G = 2$. Assume $Z_i$ is i.i.d. for $i = 1, 2$ according some unimodal distribution with mode $\nu$ with positive measure on every interval. Then $((C, \theta_1, D), (C, \theta_2, D))$ is a Nash equilibrium if and only if $\theta_1 = \theta_2 \leq \nu$.*

## 4 A folk theorem for diff meta games

What are the Nash equilibria of a diff meta game on $\Gamma$? A first answer is that Nash equilibria of $\Gamma$ carry over to the diff meta game regardless of what diff function is used (assuming that at least all constant policies are available); see Proposition 16 in Appendix C.1. Any other equilibria of the diff meta game hinge on the use of the right diff function. In fact, if diff is constant and thus uninformative, the Nash equilibria of the diff meta game are exactly the Nash equilibria of $\Gamma$; see Proposition 17 in Appendix C.1. So the next question to ask is for what strategy profiles $\sigma$ *there exists* some diff function s.t. $\sigma$ is played in an equilibrium of the resulting diff meta game. The following result answers this question. In particular, a folk theorem similar to the folk theorems for infinitely repeated games (e.g., Osborne 2004, Ch. 15) and for program equilibrium (see Section 7).

**Theorem 3** (folk theorem for diff meta games). *Let $\Gamma$ be a game and $\sigma$ be a strategy profile for $\Gamma$. Let $\mathcal{A}_i \supseteq \bar{\mathcal{A}}_i$ for $i = 1, 2$. Then the following two statements are equivalent:*

1. *There is a diff function such that there is a Nash equilibrium $(\pi_1, \pi_2)$ of the diff meta game $(\Gamma, \mathrm{diff}, \mathcal{A}_1, \mathcal{A}_2)$ s.t. $(\pi_1, \pi_2)$ play $\sigma$.*
2. *The strategy profile $\sigma$ is individually rational (i.e., better than everyone's minimax payoff).*

*The result continues to hold true if we restrict attention to deterministic diff functions with $\mathrm{diff}_1 = \mathrm{diff}_2$ and $\mathrm{diff}_i(\pi_1, \pi_2) \in \{0, 1\}$ for $i = 1, 2$.*

We leave the full proof to Appendix C.2, but give a short sketch of the construction for 2⇒1 here. For any $\sigma$, we construct the desired equilibrium from policies $\pi_i^* = (\sigma_i, 1/2, \hat{\sigma}_i)$ for $i = 1, 2$, where $\hat{\sigma}_i$ is Player $i$'s minimax strategy against Player $-i$. We then take any diff function s.t. $\mathrm{diff}(\pi_i^*, \pi_{-i}) = (0, 0)$ if $\pi_{-i} = \pi_{-i}^*$ and $\mathrm{diff}(\pi_i^*, \pi_{-i}) = (1, 1)$ otherwise.

## 5 A uniqueness theorem

Theorem 3 allows for highly asymmetric similarity-based cooperation. For example, in the PD with, say, $G = 2$, Theorem 3 shows that with the right diff function, the strategy profile $(C, 2/3*C + 1/3*D)$ is played in an equilibrium of the diff meta game of the PD. This seems odd, as one would expect SBC to result in playing symmetric strategy profiles. Note that, for example, all equilibria of Propositions 1 and 2 are symmetric. In this section, we show that under some restrictions on diff and the base game $\Gamma$, we can recover the symmetry intuition. This is good because in symmetric games the symmetric outcomes are the fair and otherwise desirable ones (Harsanyi et al., 1988, Sect. 3.4) and because SBC thus avoids equilibrium selection problems of other forms of cooperation (including cooperation based on full mutual transparency and cooperation in the iterated Prisoner's Dilemma).

We first need a few definitions of properties of $\mathrm{diff}$. Let $\Gamma$ be a symmetric game. We say that $\mathrm{diff}$ is *minimized by copies* if for all policies $\pi, \pi'$, all $y$ and $i = 1, 2$, $P(\mathrm{diff}_i(\pi, \pi') < y) \leq P(\mathrm{diff}_i(\pi, \pi) < y)$. For example, the $\mathrm{diff}$ function in Example 1 is minimized by copies. The $\mathrm{diff}$ functions in the proof of Theorem 3 are not in general minimized by copies when the given base game is symmetric. For example, to achieve $(C, {}^2\!/3 * C + {}^1\!/3 * D)$ in equilibrium, the proof of Theorem 3 (as sketched above) uses the policies $\pi_1^* = (C, {}^1\!/2, D)$ and $\pi_2^* = ({}^2\!/3 * C + {}^1\!/3 * D, {}^1\!/2, D)$ and a diff function with $\mathrm{diff}(\pi_1^*, \pi_2^*) = (0, 0)$ but $\mathrm{diff}(\pi_1^*, \pi_1^*) = (1, 1)$. If the base game is symmetric, we call $\mathrm{diff}$ *symmetric* if for all $\pi_1, \pi_2$, $\mathrm{diff}(\pi_1, \pi_2)$ is distributed the same as $\mathrm{diff}(\pi_2, \pi_1)$ and $(\mathrm{diff}_1(\pi_1, \pi_2), \mathrm{diff}_2(\pi_1, \pi_2))$ is distributed the same as $(\mathrm{diff}_2(\pi_1, \pi_2), \mathrm{diff}_1(\pi_1, \pi_2))$.

Finally, we need a more complicated but nonetheless intuitive property of $\mathrm{diff}$ functions. In this paper, we generally imagine that *low* values of $\mathrm{diff}$ are informative about the other player's policy. In contrast, we will her assume that *high* values of $\mathrm{diff}$ are uninformative. That is, for any $\sigma_i$ and $\pi_{-i}$, we will assume that there is a policy $\pi_i$ that plays $\sigma_i$ against $\pi_{-i}$ and triggers the above-threshold policy of $\pi_{-i}$ with the highest-possible probability. Formally, let $\pi_{-i} = (\sigma_{-i}^{\leqslant}, \theta_{-i}, \sigma_{-i}^{>})$ be any threshold policy. Let $p$ be the supremum of numbers $p'$ for which there is $\pi_i$ s.t. in $(\pi_i, \pi_{-i})$, Player $-i$ plays $(1 - p')\sigma_{-i}^{\leqslant} + p'\sigma_{-i}^{>}$. Let $\sigma_{\pi_{-i}}^{\max} = (1 - p)\sigma_{-i}^{\leqslant} + p\sigma_{-i}^{>}$. Intuitively, $\sigma_{\pi_{-i}}^{\max}$ is the strategy played by $\pi_{-i}$ against the most different opponent policies. For the examples of Section 3 we have $p = 1$ and thus simply $\sigma_{\pi_{-i}}^{\max} = \sigma_{-i}^{>}$. But if $\mathrm{diff}$ is bounded, then we might even have $p = 0$ or anything in between.

**Definition 3.** *We call* $\mathrm{diff} \colon \bar{\mathcal{A}}_1 \times \bar{\mathcal{A}}_2 \rightsquigarrow \mathbb{R}^2$ *high value uninformative if for each threshold policy* $\pi_{-i}$, $\sigma_i$ *and* $\epsilon > 0$ *there is a threshold policy* $\pi_i$ *such that in* $(\pi_i, \pi_{-i})$, *a strategy profile within* $\epsilon$ *of* $(\sigma_i, \sigma_{\pi_{-i}}^{\max})$ *is played.*

We are now ready to state a uniqueness result for the Nash equilibria of diff meta games.

**Theorem 4.** *Let* $\Gamma$ *be a player-symmetric, additively decomposable game. Let* $\mathrm{diff}$ *be symmetric, high-value uninformative, and minimized by copies. Then if* $(\pi_1, \pi_2)$ *is a Nash equilibrium that is not Pareto-dominated by another Nash equilibrium, we have that* $V_1(\pi_1, \pi_2) = V_2(\pi_1, \pi_2)$. *Hence, if there exists a Pareto-optimal Nash equilibrium, its payoffs are unique, Pareto-dominant among Nash equilibria and equal across the two players.*

We prove Theorem 4 in Appendix D.3. Roughly, we prove that under the given assumptions, equilibrium policies are more beneficial to the opponent when observing a diff value below the threshold than if they observe a diff value above the threshold. Second, we show that if in a given strategy profile Principal $i$ receives a lower utility than Principal $-i$, then Principal $i$ can increase her utility by submitting a copy of Principal $-i$'s policy. Appendix D.1 shows why the assumptions (additive decomposability of the game and and high-value uninformativeness and symmetry of $\mathrm{diff}$) are necessary.

## 6 Machine learning for similarity-based cooperation in complex games

Our results so far demonstrate the theoretical viability of similarity-based cooperation, but leave open questions regarding its practicality. In complex environments, where cooperating and defecting are by themselves complex operations, can we find the cooperative equilibria for a given $\mathrm{diff}$ function with machine learning methods?

### 6.1 A novel pretraining method for similarity-based cooperation

We now describe *Cooperate against Copies and Defect against Random (CCDR)*, a simple ML method to find cooperative equilibria in complex games. To use this method, we consider neural net policies $\pi_\theta$ parameterized by a real vector $\theta$. First, for any given diff game, let $V^d \colon (\mathbb{R}^m)^{(\mathbb{R}^{n+1})} \times (\mathbb{R}^m)^{(\mathbb{R}^{n+1})} \to \mathbb{R}^2$ be the utility of a version of the game in which $\mathrm{diff}$ is non-noisy. CCDR trains a model $\pi_{\theta_i}$ to maximize $V^d(\pi_{\theta_i}, \pi_{\theta_i}) + V^d(\pi_{\theta_i}, \pi_{\theta'_{-i}})$ for randomly sampled $\theta'_{-i}$. That is, each player $i$ pretrains their policy $\pi_{\theta_i}$ to do well in both of the following scenarios: principal $-i$ copies principal $i$'s model; and principal $-i$ generates a random model. The method is named for its intended effect in Prisoner's Dilemma-like games. Note, however, that it is well-defined in all symmetric games, not just Prisoner's Dilemma-like games.

CCDR pretraining is motivated by two considerations. First, in games like the Prisoner's Dilemma, there exist cooperative equilibria of policies that cooperate at a diff value of $0$ and defect as the

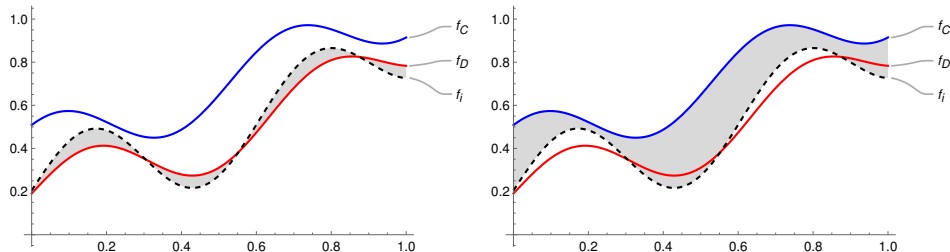

Figure 2: The figure illustrates how utilities are calculated in the HDPD. The function $f_i$ is an action chosen by Player $i$. The area between the curves $f_i$ and $f_D$ determines, intuitively, how much the agent defects; the area between the curves $f_i$ and $f_C$ determines how much it cooperates. The $f_i$ shown in the figure is much closer to defecting than to cooperation.

perceived diff value increases. We give a toy model of this in Appendix E. CCDR puts in place the rudimentary structure of these equilibria. Note, however, that CCDR does not directly optimize for the model's ability to form an equilibrium. Second, CCDR can be thought of as a form of curriculum training. Before trying to play diff games against other (different but similar) learned agents, we might first train a policy to solve two (conceptually and technically) easier related problems.

## 6.2 A high-dimensional one-shot Prisoner's Dilemma

To study similarity-based cooperation in an ML context, we need a more complex version of the Prisoner's Dilemma. The complex Prisoner's Dilemma-like games studied in the multi-agent learning community generally offer other mechanisms that establish cooperative equilibria (e.g., playing a game repeatedly). For our experiments, however, we specifically need SBC to be the only mechanism to establish cooperation.

We therefore introduce a new game, the High-Dimensional (one-shot) Prisoner's Dilemma (HDPD). The goal is to give a variant of the one-shot Prisoner's Dilemma that is conceptually simple but introduces scalable complexity that makes finding, for example, exact best responses in the diff meta game intractable. In addition to $G$, the HDPD is parameterized by two functions $f_C, f_D \colon \mathbb{R}^n \to \mathbb{R}^m$ representing the two actions Cooperate and Defect, respectively, as well as a probability measure $\mu$ over $\mathbb{R}^n$. Each player's action is also a function $f_i \colon \mathbb{R}^n \to \mathbb{R}^m$. This is illustrated in Figure 2 for the case of $n = 1$ and $m = 1$. For any pair of actions $f_1, f_2$, payoffs are then determined as follows. First, we sample some $\mathbf{x}$ according to $\mu$ from $\mathbb{R}^n$. Then to determine how much Player $i$ cooperates, we consider the distance $d(f_i(\mathbf{x}), f_C(\mathbf{x}))$ to determine, roughly speaking, how much Player 1 cooperates. The larger the distance the less cooperative is $f_i$. In the case of $n = m = 1$ and $\mu$ uniform, the expected distance between $f_i(x)$ and $f_D(x)$ is simply the area between the curves of $f_i$ and $f_D$, as visualized in Figure 2. We analogously determine how much the players defect. Formally, we define $u_i(f_1, f_2) = -\mathbb{E}_{\mathbf{x} \sim \mu}\left[d(f_i(\mathbf{x}), f_D(\mathbf{x})) + Gd(f_{-i}(\mathbf{x}), f_C(\mathbf{x}))\right] / \mathbb{E}_{\mathbf{x} \sim \mu}[d(f_C(\mathbf{x}), f_D(\mathbf{x}))]$. Thus, the action $f_i = f_D$ corresponds to defecting and the action $f_C$ corresponds to cooperating, e.g., $\mathbf{u}(f_C, f_C) = (-1, -1)$ and $\mathbf{u}(f_D, f_D) = (-G, -G)$. The unique equilibrium of this game is $(f_D, f_D)$. In our experiments, we specifically used $G = 5$.

We consider a diff meta game on the HDPD. Formally, a diff-based policy for the HDPD is a function $\mathbb{R} \to (\mathbb{R}^m)^{(\mathbb{R}^n)}$. For notational convenience, we will instead write policies as functions $\mathbb{R}^{n+1} \to \mathbb{R}^m$. We then define our diff function by $\mathrm{diff}_i(\pi_1, \pi_2) = \mathbb{E}_{(y, \mathbf{x}) \sim \nu}\left[d(\pi_1(y, \mathbf{x}), \pi_2(y, \mathbf{x}))\right] + Z_i$, where $\nu$ is some probability distribution over $\mathbb{R}^{n+1}$ and $Z_i$ is some real-valued noise.

## 6.3 Experiments

**Experimental setup.** We trained on the environment from Section 6.2. We selected a fixed set of hyperparameters based on prior exploratory experiments and the theoretical considerations in Appendix E. We then randomly initialized $\theta_1$ and $\theta_2$, CCDR-pretrained them (independently), and then trained $\theta_1$ and $\theta_2$ against each other using ABR. We repeated the experiment with 28 random seeds. As control, we also ran the experiment *without* CCDR on 26 seeds. We also ran experiments with Learning with Opponent-Learning Awareness (LOLA) (Foerster et al., 2018), which we report in Appendix G.

**Results.** First, we observe that in the runs without CCDR pretraining, the players generally converge to mutual defection during alternating best response learning. In particular, in all 26 runs, at least one

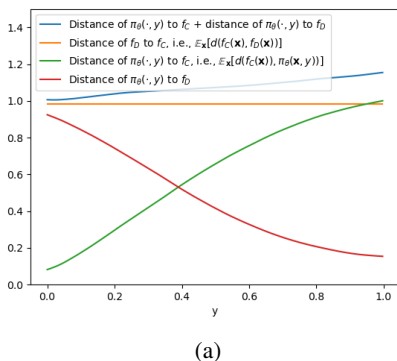

(a)

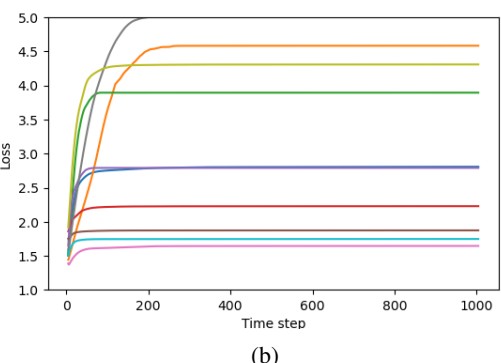

(b)

Figure 3: (a) The behavior of a CCDR-pretrained policy. For each perceived value of diff to the opponent $y$, the green line shows the expected distance of the learned policy's choice to $f_C$ (smaller means more cooperative) and the read line shows the expected distance to $f_D$. (b) Losses of Player 1 in 10 runs through the ABR phase.

player's utility was below $-5$. Only two runs had a utility above $-5$ for one of the players ($-4.997$ and $-4.554$). The average utility across the 26 runs and across the two players was $-5.257$ with a standard deviation of $0.1978$. Anecdotally, these results are robust – ABR without pretraining practically never finds cooperative equilibria in the HDPD.

Second, we observe that in all 28 runs, CCDR pretraining qualitatively yields the desired policy models, i.e., a policy that cooperates at low values of diff and gradually comes closer to defecting at high values of diff. Figure 3a shows a representative example.

Our main positive experimental result is that after CCDR pretraining, the models converged in alternating best response learning to a partially cooperative equilibrium in 26 out of 28 runs. Thus, the cooperative equilibria postulated in general by Theorem 3 and in simplified examples by Propositions 1 and 2 (as well as Proposition 25), do indeed exist and can be found with simple methods. The minimum utility of either player across the 26 successful runs was -4.854. The average utility across all runs and the two players was about -2.77 and thus a little closer to $u(f_C, f_C) = -1$ than to $u(f_D, f_D) = -5$. The standard deviation was about 1.19. Figure 3b shows the losses (i.e., the negated utilities) across ABR learning. Generally, the policies also converge to receiving approximately the same utility (cf. Section 5). The average of the absolute differences in utility between the two players at the end of the 28 runs is about 0.04 with a standard deviation of 0.05. We see that in line with Theorem 4, we tend to learn egalitarian equilibria in this symmetric, additively decomposable setting. After alternating best response learning, the models generally have a similar structure as the model in Figure 3a, though often they cooperate only a little at low diff values. Based on prior exploratory experiments, CCDR's success is moderately robust.

### 6.4 Discussion

Without pretraining, ABR learning unsurprisingly converges to mutual defection. This is due to a bootstrapping problem. Submitting a policy of the form "cooperate with similar policies, defect against different policies" is a unique best response against itself. If the opponent model $\pi_{-i}$ is not of this form, then any policy $\pi_i$ that defects, i.e., that satisfies $\pi_i(\mathrm{diff}(\pi_i, \pi_{-i})) = f_D$, is a best response. Because $f_C$ is complex, learning a model that cooperates at all is unlikely. (Even if $f_C$ was simple, the appropriate use of the perceived diff value would still be specific and thus unlikely to be found by chance.) Similar failures to find the more complicated cooperative equilibria by default have also been observed in the iterated PD (Sandholm & Crites 1996; Foerster et al. 2018; Letcher et al. 2019) and in the open-source PD (Hutter, 2020). Opponent shaping methods have been used successfully to learn to cooperate both in the iterated Prisoner's Dilemma (Foerster et al. 2018; Letcher et al. 2019) and the open-source Prisoner's Dilemma (Hutter, 2020). Our experiments in Appendix G show that LOLA can also learn SBC, but unfortunately not as robustly as CCDR pretraining.

CCDR pretraining reliably finds models that cooperate with each other and that continue to partially cooperate with each other throughout ABR training. This shows that when given some guidance, ABR can find SBC equilibria – SBC equilibria have at least some "basin of attraction". Our experiments therefore suggest that SBC is a promising means of establishing cooperation between ML agents.

That said, CCDR has many limitations that we hope can be addressed in future work. For one, in many games the best response against a randomly generated opponents does poorly against a rational opponent. Second, our experiments show that while the two policies almost fully cooperate after CCDR pretraining, they quickly partially unlearn to cooperate in the ABR phase. We would prefer a method that preserves closer to full cooperation throughout ABR-style training. Third, while CCDR seems to often work, it can certainly fail in games in which SBC is possible. Learning to distinguish randomly sampled opponent policies from copies will in many settings not prepare an agent to distinguish cooperative/SBC opponents from uncooperative but trained (not randomly sampled) opponents. Consequently, CCDR may sometimes result in insufficiently steep incentive curves, cooperating with too dissimilar opponents. We suspect that to make progress on the latter issues we need training procedures that more explicitly reason about incentives *à la* opponent shaping (cf. our experiments with LOLA Appendix G).

## 7 Related work

We here relate our project to the two most closely related lines of work. In Appendix H we discuss more distantly related lines of work.

**Program equilibrium.** We already discussed in Section 1 the literature on program meta games in which players submit computer programs as policies and the programs fully observe each other's code (McAfee 1984; Howard 1988; Rubinstein 1998, Section 10.4; Tennenholtz 2004). Interestingly, some constructions for equilibria in program meta games are similarity based. For example, the earliest cooperative program equilibrium for the Prisoner's Dilemma, described in all four of the above-cited papers, is the program "Cooperate if the opponent's program is equal to this program; else Defect". The program "cooperate if my cooperation implies cooperation from the opponent" proposed by Critch et al. (2022) is also similarity-based. Other approaches to program equilibrium cannot be interpreted as similarity based, however (see, e.g., Barasz et al., 2014; Critch, 2019; Oesterheld, 2019b). To our knowledge, the only published work on ML in program equilibrium is due to Hutter (2020). It assumes the programs to have the structure proposed by Oesterheld (2019b) on simple normal-form games, thus leaving only a few parameters open. Similar to our experiments, Hutter shows that best response learning fails to converge to the cooperative equilibria. In Hutter's experiments, the opponent shaping methods LOLA (Foerster et al., 2018) and SOS (Letcher et al., 2019) converge to mutual cooperation.

**Decision theory and Newcomb's problem.** Brams (1975) and Lewis (1979) have pointed out that the Prisoner's Dilemma against a similar opponent closely resembles *Newcomb's problem*, a problem first introduced to the decision-theoretical literature by Nozick (1969). Most of the literature on Newcomb's problem is about the normative, philosophical question of whether one should cooperate or defect in a Prisoner's Dilemma against an exact copy. Our work is inspired by the idea that in some circumstances one should cooperate with similar opponents. However, this literature only informally discusses the question of whether to also cooperate with agents other than exact copies (Hofstadter e.g., 1983; Drescher 2006, Ch. 7; Ahmed 2014, Sect. 4.6.3). We address this question formally.

One idea behind the present project, as well as the program game literature, is to analyze a decision situation from the perspective of (actual or hypothetical) *principals* who design policies. The principals find themselves in an ordinary strategic situation. This is how our analysis avoids the philosophical issues arising in the *agent's* perspective. Similar changes in perspective have been discussed in the literature on Newcomb's problem (e.g., Gauthier 1989; Oesterheld & Conitzer 2022).

## 8 Conclusion and future work

We make a strong case for the promise of similarity-based cooperation as a means of improving outcomes from interactions between ML agents. At the same time, there are many avenues for future work. On the theoretical side, we would be especially interested in generalizations of Theorem 4, that is, theorems that tell us what outcomes we should expect in diff meta games. Is it true more generally that under reasonable assumptions about the diff function, we can expect SBC to result in fairly specific, symmetric, Pareto-optimal outcomes? We are also interested in further experimental investigations of SBC. We hope that future work can improve on our results in the HDPD in terms of robustness and degree of cooperation. Besides that, we think a natural next step is to study settings in which the agents observe their similarity to one another in a more realistic fashion. For example, we conjecture that SBC can occur when the agents can determine that their policies were generated by similar learning procedures.

## Acknowledgments

We thank Stephen McAleer, Emery Cooper, Daniel Filan, John Mori, and our anonymous reviewers helpful discussions and comments. We thank Maxime Riché and the Center on Long-Term Risk for compute support. Caspar Oesterheld and Vincent Conitzer would like to thank the Cooperative AI Foundation, Polaris Ventures (formerly the Center for Emerging Risk Research) and Jaan Tallinn's donor-advised fund at Founders Pledge for financial support. Roger Grosse acknowledges financial support from Open Philanthropy. Caspar Oesterheld and Johannes Treutlein are grateful for support by FLI PhD Fellowships. Johannes Treutlein was additionally supported by an OpenPhil AI PhD Fellowship.

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

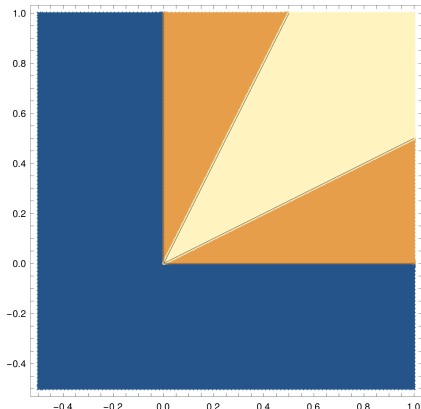

Figure 4: Visualization of outcomes as a function of the thresholds in Example 1 without noise $(Z_1 = Z_2 = 0)$. For each pair of thresholds (x and y axis), the graph shows whether both players cooperate (yellow), both players defect (blue), or one player cooperates and the other defects (orange).

## A   Preliminary game theory results

We say that $\boldsymbol{\sigma}$ *very weakly Pareto-dominates* $\hat{\boldsymbol{\sigma}}$ if for all $i$, we have that $u_i(\boldsymbol{\sigma}) \geq u_i(\hat{\boldsymbol{\sigma}})$.

**Proposition 5.** *Let $\Gamma$ be a two-player additively decomposable normal-form game. Then $\Gamma$ has a Nash equilibrium that very weakly Pareto-dominates all other Nash equilibria. If $\Gamma$ is furthermore symmetric, then in the Pareto-dominant equilibrium, both players receive the same utility.*

*Proof.* First note that for all $\sigma_{-i}$, Player $i$'s best responses are given by

$$\arg\max_{a_i} u_i(a_i, \sigma_{-i}) = \arg\max_{a_i} u_{i,i}(a_i) + u_{i,-i}(\sigma_{-i}) = \arg\max_{a_i} u_{i,i}(a_i).$$

Now among this set of universal best responses, let let $a_i^*$ be one that maximizes $u_{-i,i}(a_i)$ for $i = 1, 2$. Clearly $\mathbf{a}^*$ is a Nash equilibrium.

Now let $\boldsymbol{\sigma}$ be any Nash equilibrium. Note that for $i = 1, 2$ the support of $\sigma_i$ must be in the above argmax. It follows that for $i = 1, 2$,

$$u_i(\mathbf{a}^*) = u_{i,i}(a_i^*) + u_{i,-i}(a_{-i}^*) = u_{i,i}(\sigma_i) + u_{i,-i}(a_{-i}^*) \geq u_{i,i}(\sigma_i) + u_{i,-i}(\sigma_{-i}) = u_i(\boldsymbol{\sigma}).$$

$\square$

## B   A detailed analysis of Example 1

Example 1 is already surprisingly rich. We here provide a detailed analysis.

**Example 1.** *Let $\Gamma$ be the Prisoner's Dilemma as per Table 1. Then consider the $(\Gamma, \hat{A}_1, \hat{A}_2, \text{diff})$ meta game where $\hat{A}_i = \{(C, \theta_i, D) \mid \theta_i \in \mathbb{R}\}$ and $\text{diff}_i((C, \theta_1, D), (C, \theta_2, D))) = |\theta_1 - \theta_2| + Z_i$ for $i = 1, 2$ where $Z_i$ is some real-valued random variable.*

Figures 4, 5a and 5b illustrate Example 1. Specifically, Figure 4 considers the case without noise $(Z_i = 0)$ and shows for each pair of thresholds $\theta_1, \theta_2$ whether both agents cooperate (yellow), only one agent (the one with the lower threshold) cooperates (orange), or both defect (blue). Figure 5a and Figure 5b consider the case $Z_i \sim \text{Uniform}([0, 1])$, i.e., the case where noise is drawn uniformly from $[0, 1]$. Figure 5a shows for each pair of thresholds the minimum probability of cooperation across the two players. For instance, if Player 1 submits $0.5$ and Player 2 submits $1$, then Player 1 cooperates with probability $0$ and and Player 2 cooperates with probability $1/2$, so the plot in Figure 5a is $0$ (blue) at $(0.5, 1)$ (and symmetrically at $(1, 0.5)$). Figure 5b shows the action of the agent whose threshold is given by the x axis.

Because we here restrict attention to policies of type $(C, \theta, D)$, policies are uniquely specified by a single real number $\theta$. So we will denote them as such.

In addition to threshold policies that correspond to real numbers, we will here consider the agents $-\infty$ by which we mean the agent that always defects, and the agent $\infty$, by which we mean the agent that always cooperates.

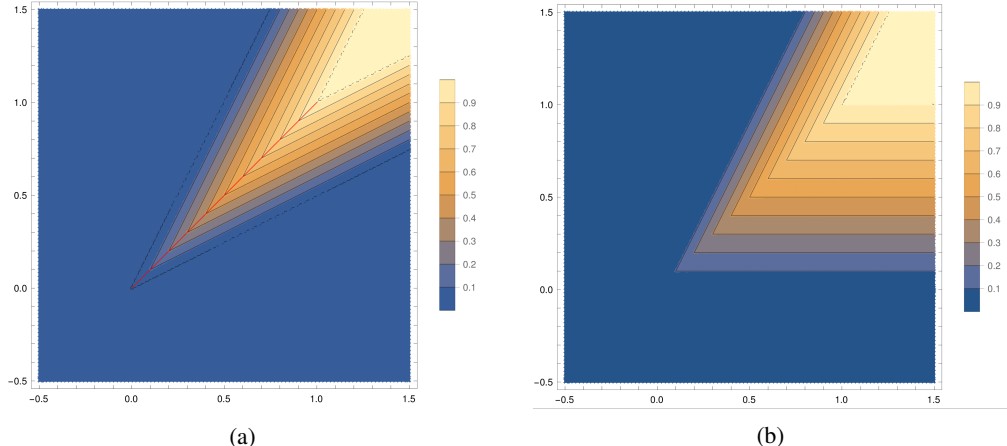

Figure 5: Visualization of the probabilities of cooperation as a function of the thresholds in Example 1 with uniform noise $Z_1, Z_2 \sim \mathrm{Uniform}([0,1])$. For each pair of thresholds (x and y axis), the left graph shows the minimum probability of cooperation across the two players. The right-hand graph shows the probability of cooperation of the player corresponding to the x axis.

One might suspect that if there is too much noise, there can be no cooperative equilibria. But it's easy to see that the setting of Example 1 is scale-invariant.

**Proposition 6** (Scale invariance of noise)**.** *Let* $(\Gamma, \mathcal{A}_1, \mathcal{A}_2, \mathrm{diff})$ *be a* diff-*based meta game with utility* $V$, *where* $\mathrm{diff}(\theta_1, \theta_2) = |\theta_1 - \theta_2| + Z_i$. *Further, let* $(\Gamma, \mathcal{A}_1, \mathcal{A}_2, \mathrm{diff}')$ *be a* diff-*based meta game with utility* $V'$, *where* $\mathrm{diff}'(\theta_1, \theta_2) = |\theta_1 - \theta_2| + \alpha Z_i$ *for some* $\alpha > 0$. *Then for all* $\theta_1, \theta_2$, $V(\theta_1, \theta_2) = V'(\alpha\theta_1, \alpha\theta_2)$. *It follows that for all* $\theta_1, \theta_2$, $(\theta_1, \theta_2)$ *is a Nash equilibrium in the* diff *meta game if and only if* $(\alpha\theta_1, \alpha\theta_2)$ *is a Nash equilibrium in the* diff' *meta game.*

### B.1   Best responses

In the regular Prisoner's Dilemma, defecting strictly dominates cooperating. Similarly, in the diff meta game of Example 1, always defecting strictly dominates always cooperating (without looking at the difference to the opponent).

**Definition 4.** *Let* $(A_1, A_2, \mathbf{u})$ *be a normal-form game. Let* $a_1, a_1' \in A_1$ *be strategies for Player 1. We say that* $a_1$ *very weakly dominates* $a_1'$ *if for all* $a_2 \in A_2$ *we have that* $u_1(a_1, a_2) \geq u_1(a_1, a_2)$. *We further say that* $a_1$ *weakly dominates* $a_1'$ *if the inequality is strict for at least one* $a_2$ *and that* $a_1$ *strictly dominates* $a_1$ *if the inequality is strict for all* $a_2$.

**Proposition 7.** *The threshold policy* $-\infty$ *strictly dominates the threshold policy* $\infty$.

Intuitively, in our model there is never a reason submit a policy that defects when it can be sure that it faces an exact copy. If the noise is lower-bounded, this puts a lower bound on what kind of agent it makes sense to submit, as we now show.

**Proposition 8.** *Let* $Z_i \geq \theta_i$ *with certainty. Let* $\theta_i' < \theta_i$. *Then* $\theta_i$ *very weakly dominates* $\theta_i'$. *If* $P(Z = \theta_i) > 0$, $\theta_i$ *weakly dominates* $\theta_i'$.

The next result shows that if the player who submits the higher threshold decreases her threshold while still staying above the other player's threshold, she cooperates with the same probability. Conversely, one cannot (in the setting of Example 1) decrease one's probability of threshold by increasing one's threshold.

**Lemma 9.** *Let* $\theta_i, \theta_i', \theta_{-i} \in \mathbb{R}$ *with* $\theta_i \geq \theta_i' \geq \theta_{-i}$. *Then in* $(\theta_i, \theta_{-i})$, *Player* $i$ *cooperates with equal probability as in* $(\theta_i', \theta_{-i})$.

*Proof.*

$$
\begin{aligned}
P(\text{Pl. } i \text{ C's} \mid \theta_i, \theta_{-i}) &= P\left(Z_i + \theta_i - \theta_{-i} \leq \theta_i\right) \\
&= P\left(Z \leq \theta_{-i}\right) \\
&= P\left(Z_i + \theta_i' - \theta_{-i} \leq \theta_i'\right) \\
&= P(\text{Pl. } i \text{ C's} \mid \theta_i', \theta_{-i})
\end{aligned}
$$

$\square$

**Theorem 10.** *Let $\theta_i, \theta_{-i} \in \mathbb{R}$ with $\theta_i > \theta_{-i}$. Then $u_i(\theta_{-i}, \theta_{-i}) \geq u_i(\theta_i, \theta_{-i})$. The inequality is strict if and only if $P(2\theta_{-i} - \theta_i \leq Z_{-i} \leq \theta_{-i}) > 0$.*

Intuitively, there's never a reason to submit a higher threshold than the opponent.

*Proof.* With Lemma 9, we need only prove that by decreasing $\theta_i$ to $\theta_{-i}$ the probability that Player 2 cooperates (weakly) increases. This is easy to see, though for the strictness condition, we need the details:

$$
\begin{aligned}
P(\text{Pl. } -i \text{ C's} \mid \theta_i, \theta_{-i}) &= P\left(Z_{-i} + \theta_i - \theta_{-i} \leq \theta_{-i}\right) \\
&= P\left(Z_{-i} \leq 2\theta_{-i} - \theta_i\right) \\
P(\text{Pl. } -i \text{ C's} \mid \theta_{-i}, \theta_{-i}) &= P\left(Z_{-i} \leq \theta_{-i}\right)
\end{aligned}
$$

Clearly, $P\left(Z_{-i} \leq \theta_{-i}\right) \geq P\left(Z_{-i} \leq 2\theta_{-i} - \theta_i\right)$. Moreover, the inequality is strict if and only if $P(2\theta_{-i} - \theta_i \leq Z_{-i} \leq \theta_{-i}) > 0$. $\square$

### B.2 (Pure) Nash equilibria

We now give some results on the Nash equilibria of Example 1. We start with two simple results to warm up.

**Proposition 11.** *For all distributions of the noise:*

1. $(-\infty, -\infty)$ *is a Nash equilibrium.*

2. $(\infty, \infty)$ *is not a Nash equilibrium.*

**Proposition 12.** *If there is no upper bound to noise, then there is no fully cooperative equilibrium.*

*Proof.* If there is no upper bound to noise, then the only policy profile with universal cooperation is $(\infty, \infty)$. But by Proposition 11.2, this is not a Nash equilibrium. $\square$

Next we use our results on best responses to show that to form a Nash equilibrium it is never *necessary* for the two players to submit different thresholds.

**Theorem 13.** *Let $(\theta_i, \theta_{-i})$ be a Nash equilibrium with $\theta_i > \theta_{-i}$. Then $(\theta_{-i}, \theta_{-i})$ is also a Nash equilibrium.*

*Proof.* WLOG assume $i = 1$ for notational clarity. Assume for contradiction that $(\theta_2, \theta_2)$ is not a Nash equilibrium. First, notice that by Theorem 10 and the assumption that $\theta_1$ is a best response for Player 1 to $\theta_2$, it follows that for Player 1 $\theta_2$ is a best response to $\theta_2$. So if $(\theta_2, \theta_2)$ is not a Nash equilibrium, then it must be because for Player 2 $\theta_2$ is not a best response to $\theta_2$ as submitted by Player 1. So there must be $\theta_2'$ such that $u_2(\theta_2, \theta_2') > u_2(\theta_2, \theta_2)$. By Theorem 10, $\theta_2' < \theta_2$.

We now show that we would then also have that $u_2(\theta_1, \theta_2') > u_2(\theta_1, \theta_2)$ in contradiction with the assumption that $(\theta_1, \theta_2)$ is a Nash equilibrium. We do this via the following sequence of (in)equalities:

$$
u_2(\theta_1, \theta_2') \underset{(1)}{\geq} u_2(\theta_2, \theta_2') > u_2(\theta_2, \theta_2) \underset{(2)}{=} u_2(\theta_1, \theta_2).
$$

(1) By Lemma 9, Player 1 cooperates with equal probability in $(\theta_1, \theta_2')$ and $(\theta_2, \theta_2')$. It is easy to see that Player 2's probability of cooperating is weakly lower in $(\theta_1, \theta_2')$. It follows that $u_2(\theta_1, \theta_2') \geq u_2(\theta_2, \theta_2')$.

(2) (A) By Lemma 9, Player 1 cooperates with equal probability in $(\theta_1, \theta_2)$ and $(\theta_2, \theta_2)$. (B) From A and the assumption that $\theta_1 \in \text{BR}_1(\theta_2)$ it follows that Player 2 cooperates with equal probability in $(\theta_1, \theta_2)$ and $(\theta_2, \theta_2)$. (Because if this were not the case, then $\theta_2$ would be a strictly better response for Player 1 to $\theta_2$.) From A and B it follows that the distributions over actions are the same in $(\theta_1, \theta_2)$ and $(\theta_2, \theta_2)$ and thus that $u_2(\theta_2, \theta_2) = u_2(\theta_1, \theta_2)$ as claimed. $\square$

We are now ready to show the first of our two results about the main text.

**Proposition 1.** *Consider Example 1 with $Z_i \sim \mathrm{Uniform}([0, \epsilon])$ i.i.d. for some $\epsilon > 0$ and with $G \geq 2$. Then $((C, \theta_1, D), (C, \theta_2, D))$ is a Nash equilibrium if and only if $\theta_1, \theta_2 \leq 0$ or $0 < \theta_1 = \theta_2 \leq \epsilon$. In case of the latter, the equilibrium is strict if $G > 2$.*

*Proof.* "$\Leftarrow$": First we show that the given strategy profiles really are equilibria.

1. $\theta_1, \theta_2 \leq 0$ is just the like the earlier $(-\infty, -\infty)$ equilibrium. If one player plays $\theta_{-i} \leq 0$, then clearly the unique best response is to also always defect.

2. By Theorem 10 we only need to consider whether one of the players, WLOG Player 1, can increase her utility by *decreasing* their threshold. So for the following consider $\theta_1 < \theta_2$

$$
\begin{aligned}
P(\text{Pl. 1 C's} \mid \theta_1, \theta_2) &= P(\theta_2 - \theta_1 + Z_1 < \theta_1) \\
&= P(Z_1 < 2\theta_1 - \theta_2).
\end{aligned}
$$

This is equal to $\max(0, (2\theta_1 - \theta_2)/\epsilon)$. Clearly, if Player 1 can profitably deviate to some $\theta_1$, then she can profitably deviate to some $\theta_1$ s.t. $(2\theta_1 - \theta_2)/\epsilon$ is nonnegative. After all, Player 1 wants to maximize Player 2's probability of cooperation.

Similarly,

$$
\begin{aligned}
P(\text{Pl. 2 C's} \mid \theta_1, \theta_2) &= P(\theta_2 - \theta_1 + Z < \theta_2) \\
&= P(Z < \theta_1) \\
&= \theta_1/\epsilon.
\end{aligned}
$$

Now $\theta_1 = \theta_2$ is a best response to $\theta_2$ if and only if the rate at which $P(\text{Pl. 1 C's} \mid \theta_1, \theta_2)$ decreases is at most $G$ times as high as the rate at which $P(\text{Pl. 2 C's} \mid \theta_1, \theta_2)$ decreases. Now the rates of change / derivatives are $2/\epsilon$ and $1/\epsilon$. So this condition is satisfied (for our payoff matrix).

"$\Rightarrow$": It is left to show that no other profile is a Nash equilibrium.

First, notice that for all $\theta_{-i} > \epsilon$, the unique best response is $\theta_i = \epsilon$, which minimizes the probability of $i$ cooperating, while ensuring that Player $-i$ cooperates with probability 1. For this, use part 2 of "$\Leftarrow$". From this it follows directly that there is no equilibrium in which both players play $> \epsilon$. By the strictness part of $\Leftarrow$, all equilibria in which one player plays $\leq \epsilon$ are as described in the result. $\square$

We now prove a lemma in preparation for proving our second result for the main text.

**Lemma 14.** *Assume $G = 2$ and assume that the two players have the same noise distribution. Then $(\theta, \theta)$ is a Nash equilibrium if and only if for all $\Delta > 0$, $P(\theta - 2\Delta < Z < \theta - \Delta) \leq P(\theta - \Delta < Z < \theta)$. It is a strict Nash equilibrium if all of these inequalities are strict.*

*Proof.* By Theorem 10 we only need to consider deviations to a lower threshold. So consider WLOG the case where Player 1 deviates from $\theta$ to submit $\theta - \Delta$. First, we calculate the probabilities of cooperation under $(\theta, \theta)$ and $(\theta - \Delta, \theta)$:

$$
\begin{aligned}
P(\text{Pl. 1 C's} \mid \theta, \theta) &= P(Z \leq \theta) \\
P(\text{Pl. 1 C's} \mid \theta - \Delta, \theta) &= P(Z + \Delta \leq \theta - \Delta) \\
&= P(Z \leq \theta - 2\Delta) \\
P(\text{Pl. 2 C's} \mid \theta, \theta) &= P(Z \leq \theta) \\
P(\text{Pl. 2 C's} \mid \theta - \Delta, \theta) &= P(Z + \Delta \leq \theta) \\
&= P(Z \leq \theta - \Delta)
\end{aligned}
$$

Thus by Player 1 switching from $\theta$ to $\theta - \Delta$, Player 1's probability of cooperating decreases by

$$
P(Z \leq \theta) - P(Z \leq \theta - 2\Delta) = P(\theta - 2\Delta \leq Z \leq \theta).
$$

Meanwhile, Player 2's probability of cooperating decreases by

$$
P(Z \leq \theta) - P(Z \leq \theta - \Delta) = P(\theta - \Delta \leq Z \leq \theta).
$$

Thus, for this switch to not be profitable for player 1, it needs to be the case that

$$
P(\theta - 2\Delta \leq Z \leq \theta) \leq 2P(\theta - \Delta \leq Z \leq \theta),
$$

or, equivalently,

$$P(\theta - 2\Delta \leq Z \leq \theta - \Delta) \leq P(\theta - \Delta \leq Z \leq \theta)$$

as claimed. □

**Proposition 2.** *Consider Example 1 with $G = 2$. Assume $Z_i$ is i.i.d. for $i = 1, 2$ according some uni-modal distribution with mode $\nu$ with positive measure on every interval. Then $((C, \theta_1, D), (C, \theta_2, D))$ is a Nash equilibrium if and only if $\theta_1 = \theta_2 \leq \nu$.*

*Proof.* By Theorem 10 and the assumption of positive measure on any interval, all Nash equilibria have the form $\theta_1 = \theta_2$. The second part follows directly from Lemma 14 and the fact that the noise distribution is unimodal with mode $\nu$. □

### B.3 A different type of noise

Intuitively, we might expect that more noise is an obstacle to similarity-based cooperation. The above results do not vindicate this intuition (see Proposition 6). We here give an alternative setup with a different kind of noise in which more noise *is* an obstacle to cooperation.

**Example 2.** *Consider a variant of Example 1 where for $i = 1, 2$ we have with probability $p_i$ that $\mathrm{diff}_i((C, \theta_1, D), (C, \theta_2, D))) = |\theta_1 - \theta_2| + Z_i$ with $Z_i \sim \mathrm{Unif}([0, \epsilon])$ for some $\epsilon > 0$; and with the remaining probability $\mathrm{diff}_i((C, \theta_1, D), (C, \theta_2, D))) = 0$.*

Note that for $p_1 = p_2 = 1$ the setting is exactly the setting of Proposition 1.

Intuitively, this models a scenario in which each player can try to manipulate the diff value to $0$ and the manipulation succeeds with probability $1 - p_i$. (It is further implicitly assumed, that if manipulation fails, the other player never learns of the attempt to manipulate. Instead, the diff value is observed normally if manipulation fails. That way we can assume that each player always attempts to manipulate.)

We can generalize Proposition 1 to this new setting as follows:

**Proposition 15.** *In Example 2, $((C, \theta_1, D), (C, \theta_2, D))$ is a Nash equilibrium if and only if*

- $\theta_1, \theta_2 \leq 0$; *or*
- $0 < \theta_1 = \theta_2 \leq \epsilon$ *and* $Gp_i \geq 0$ *for* $i = 1, 2$.

The proof works the same as the proof of Proposition 1.

## C Proofs for Section 4

### C.1 Nash equilibria of the base game as Nash equilibria of the meta game

We first note two simple results. The first is that every Nash equilibrium of the base game is also a Nash equilibrium of the diff meta game in which both players submit a policy that simply ignores the diff value.

**Proposition 16.** *Let $\Gamma$ be a game and $\boldsymbol{\sigma}$ be a Nash equilibrium of $\Gamma$. For $i = 1, 2$, let $\mathcal{A}_i$ be any set of policies that contains the policy $\pi_i \colon d \mapsto \sigma_i$. Then $(\pi_1, \pi_2)$ is a Nash equilibrium of the $(\Gamma, \mathrm{diff}, \mathcal{A}_1, \mathcal{A}_2)$ meta game.*

If the $\mathrm{diff}$ function is uninformative, then the Nash equilibria are in fact the only Nash equilibria of the diff meta game, as we now state.

**Proposition 17.** *Let $\Gamma$ be a game and $(\Gamma, \mathrm{diff}, \mathcal{A}_1, \mathcal{A}_2)$ be a meta game on $\Gamma$ where $\mathrm{diff}(\cdot, \cdot) = \mathbf{y}$ for some $\mathbf{y}^2$. Then $(\pi_1, \pi_2)$ is a Nash equilibrium of the meta game if and only if $(\pi_1(y_1), \pi_2(y_2))$ is a Nash equilibrium of $\Gamma$.*

### C.2 Proof of Theorem 3

**Theorem 3** (folk theorem for diff meta games). *Let $\Gamma$ be a game and $\boldsymbol{\sigma}$ be a strategy profile for $\Gamma$. Let $\mathcal{A}_i \supseteq \bar{\mathcal{A}}_i$ for $i = 1, 2$. Then the following two statements are equivalent:*

1. *There is a $\mathrm{diff}$ function such that there is a Nash equilibrium $(\pi_1, \pi_2)$ of the diff meta game $(\Gamma, \mathrm{diff}, \mathcal{A}_1, \mathcal{A}_2)$ s.t. $(\pi_1, \pi_2)$ play $\boldsymbol{\sigma}$.*
2. *The strategy profile $\boldsymbol{\sigma}$ is individually rational (i.e., better than everyone's minimax payoff).*

<table>
<table>
<tr><td></td><td></td><td colspan="3" align="center">Player 2</td></tr>
<tr><td></td><td></td><td align="center">$a_0$</td><td align="center">$a_1$</td><td align="center">$a_2$</td></tr>
<tr><td></td><td align="center">$a_0$</td><td align="center">$0, 0$</td><td align="center">$-2, 1$</td><td align="center">$-5, -1$</td></tr>
<tr><td align="center">Player 1</td><td align="center">$a_1$</td><td align="center">$1, -2$</td><td align="center">$-1, -1$</td><td align="center">$-4, -3$</td></tr>
<tr><td></td><td align="center">$a_2$</td><td align="center">$-1, -5$</td><td align="center">$-3, -4$</td><td align="center">$-6, -6$</td></tr>
</table>

Table 2: A game to show the need for assuming high-value uninformativeness in Theorem 4.

*The result continues to hold true if we restrict attention to deterministic* diff *functions with* $\mathrm{diff}_1 = \mathrm{diff}_2$ *and* $\mathrm{diff}_i(\pi_1, \pi_2) \in \{0, 1\}$ *for* $i = 1, 2$.

*Proof.* "1⇒2": We show the contrapositive, i.e., that if $\sigma$ is not individually rational, it is not implemented by any equilibrium of any diff game. Let $\hat{\sigma}_i$ be the strategy of Player $i$ that guarantees her her minimax utility. Then submitting $(\hat{\sigma}_i, \theta, \hat{\sigma}_i)$ for any threshold $\theta$ guarantees minimax utility regardless of what diff function is used. Thus, anything that gives $i$ less than threat point utility cannot be an equilibrium.

"2⇒1": Let $\tilde{\sigma}_i$ be Player $i$'s minimax strategy against Player $-i$. Then consider the strategy profile $(\pi_1 = (\tilde{\sigma}_1, 1/2, \sigma_1), \pi_2 = (\tilde{\sigma}_2, 1/2, \sigma_2))$ and any diff function s.t. $\mathrm{diff}(\pi_1, \pi_2) = 0$ and for $i = 1, 2$ and $\pi'_{-i} \neq \pi_{-i}$, $\mathrm{diff}(\pi_i, \pi'_{-i}) = 1$. Clearly, in $(\pi_1, \pi_2)$, the players play $(\sigma_1, \sigma_2)$. Finally, $(\pi_1, \pi_2)$ is a Nash equilibrium of the resulting diff meta game, because if either player deviates they will receive their minimax utility, which is by assumption no larger than their utility in $\sigma$ and thus in $(\pi_1, \pi_2)$. $\qquad\square$

# D  On the uniqueness theorem

## D.1   Examples to show the need for the assumptions of Theorem 4

### D.1.1   Why the diff function must be high-value uninformative in Theorem 4

We now give an example for why we need diff to be high-value uninformative, both for Lemma 22 and for our uniqueness theorem below.

**Proposition (Example) 18.** *Consider the game of Table 2. Note that the game is symmetric and additively decomposable. Consider the* diff *function defined by* $\mathrm{diff}((\sigma_1^{\leqslant}, *, \sigma_1^{>})_{i=1,2}) = 1$ *if* $\mathrm{supp}(\sigma_1^{\leqslant}) \cup \mathrm{supp}(\sigma_1^{>})$ *and* $\mathrm{supp}(\sigma_2^{\leqslant}) \cup \mathrm{supp}(\sigma_2^{>})$ *are disjoint and* $\mathrm{diff}((\sigma_1^{\leqslant}, *, \sigma_1^{>})_{i=1,2}) = 0$ *otherwise. Then* $((a_2, 1/2, a_1), (a_0, *, a_0))$ *is an equilibrium of the* diff *meta game.*

Intuitively, the policy $(a_2, 1/2, a_1)$ with the described diff function implements the following idea: "I want to play $a_1$ (which is good for me and moderately bad for you). I don't want you to also play $a_1$. If you are similar to me (which you are if you give weight to the same action I give weight to), I'll play $a_2$, which is very bad for you." Assuming that Player 1 submits such a policy, Player 2 optimizes her utility by always playing $a_0$. Player 1 thus obtains her favorite outcome.

### D.1.2   Why the game must be additively decomposable in Theorem 4

The following example shows why we need to restrict attention to additively decomposable games. Intuitively, the game is a Prisoner's Dilemma, except that if the players cooperate, they also play a Game of Chicken for an additional payoff. Then (with a natural diff function) similarity-based cooperation takes care of the cooperate versus defect part, but leaves open the Dare versus Swerve part. In particular, there are multiple Pareto-optimal equilibria.

**Proposition (Example) 19.** *Let* $\Gamma$ *be the game of Table 3. Define* $\mathrm{diff}(\pi_1, \pi_2) = 0$ *if* $a_0 \notin \mathrm{supp}(\pi_i(0))$ *for* $i = 1, 2$ *and* $\mathrm{diff}(\pi_1, \pi_2) = 1$ *otherwise. Then for* $i = 1, 2$,

$$(\pi_i = (a_1, 1/2, a_0), \pi_{-i} = (a_2, 1/2, a_0))$$

*is a Pareto-optimal Nash equilibrium of the* diff *meta game on* $\Gamma$.

| | | Player 2 | | |
|---|---|---|---|---|
| | | $a_0$ | $a_1$ | $a_2$ |
| Player 1 | $a_0$ | $0, 0$ | $5, -5$ | $5, -5$ |
| | $a_1$ | $-5, 5$ | $0, 0$ | $3, 1$ |
| | $a_2$ | $-5, 5$ | $1, 3$ | $1, 1$ |

Table 3: A game to show the need for the additive decomposability assumption in Theorem 4.

| | | Player 2 | | |
|---|---|---|---|---|
| | | $C_1$ | $C_2$ | $D$ |
| Player 1 | $C_1$ | $3, 3$ | $2, 3$ | $0, 4$ |
| | $C_2$ | $3, 2$ | $2, 2$ | $0, 3$ |
| | $D$ | $4, 0$ | $3, 0$ | $1, 1$ |

Table 4: A game to show why we need $\mathrm{diff}$ to be policy-symmetric in Theorem 4.

#### D.1.3 Why diff must be observer-symmetric for Theorem 4

We here give an example to show why the $\mathrm{diff}$ function in Theorem 4 needs to have $(\mathrm{diff}_1(\pi_1, \pi_2), \mathrm{diff}_2(\pi_1, \pi_2))$ and $(\mathrm{diff}_2(\pi_1, \pi_2), \mathrm{diff}_1(\pi_1, \pi_2))$ have the same distribution. One might call this observer symmetry.

**Proposition (Example) 20.** *Let $\Gamma$ be the Prisoner's Dilemma. Let $\mathrm{diff}_1(\pi_1, \pi_2) = 0$ if $\pi_1 = \pi_2$ and $\mathrm{diff}_1(\pi_1, \pi_2) = 1$ otherwise. Let $\mathrm{diff}_2(\pi_1, \pi_2)$ be defined in the same way, except that if $\pi_1 = \pi_2$, then there is still an $\epsilon$ probability of $\mathrm{diff}_2(\pi_1, \pi_2) = 1 - \epsilon$. Note that this diff meta game satisfies the other conditions of Theorem 4, i.e., $\Gamma$ is symmetric and additively decomposable, $\mathrm{diff}(\pi_1, \pi_2)$ and $\mathrm{diff}(\pi_2, \pi_1)$ are equally distributed for all $\pi_1, \pi_2$ and $\mathrm{diff}$ is high-value-uninformative and minimized by copies. Then $((C, 1/2, D), (C, 1/2, D))$ has asymmetric payoffs but is a Pareto-optimal Nash equilibrium of the $\mathrm{diff}$ meta game on $\Gamma$.*

In this example, the Pareto-optimal Nash equilibrium is still unique, but it is easy to come up with examples in which there are multiple Pareto-optimal Nash equilibria.

Note also that in this example the player who has less information about the other does better in the cooperative equilibria.

#### D.1.4 Why diff must be policy-symmetric for Theorem 4

Finally we give an example to show why the $\mathrm{diff}$ function in Theorem 4 needs to have $\mathrm{diff}(\pi_1, \pi_2)$ and $\mathrm{diff}(\pi_2, \pi_1)$ have the same distribution. One might call this policy symmetry.

**Proposition (Example) 21.** *Let $\Gamma$ be the game of Table 4. Let $\mathrm{diff}(\pi_1, \pi_2) = (0, 0)$ if $(\pi_1, \pi_2)$ equals one of the following*

$$((C_2, 1/2, D), (C_1, 1/2, D))$$
$$((C_1, 1/2, D), (D, 1/2, D))$$
$$((C_2, 1/2, D), (C_2, 1/2, D))$$
$$((C_1, 1/2, D), (C_1, 1/2, D))$$
$$((D, 1/2, D), (D, 1/2, D))$$

*and $\mathrm{diff}(\pi_1, \pi_2) = (1, 1)$ otherwise. Note that this diff meta game satisfies the other conditions of Theorem 4, i.e., $\Gamma$ is symmetric and additively decomposable, $\mathrm{diff}_1(\pi_1, \pi_2) = \mathrm{diff}_2(\pi_1, \pi_2)$ for all $\pi_1, \pi_2$ and $\mathrm{diff}$ is high-value-uninformative and minimized by copies. However, the only Pareto-optimal (pure) Nash equilibrium of the diff meta game is $((C_2, 1/2, D), (C_1, 1/2, D))$.*

As in the previous example, the Pareto-optimal Nash equilibrium is still unique, but it is easy to come up with examples in which there are multiple Pareto-optimal Nash equilibria.

### D.2 Results on the structure of equilibria

We have various intuitions about similarity-based cooperation. For example, we have the intuition that $(C, \theta, D)$ is a sensible policy but $(D, \theta, C)$ is not. In this section we prove results of this type

under appropriate assumptions. We find these results interesting independently, but we also need all results here to prove Theorem 4.

The following two lemmas capture the idea that under some assumptions it is rational to be more cooperative if the opponent is similar, i.e., if the observed difference is below the threshold.

**Lemma 22.** *Let $(\pi_1, \pi_2)$ be a Nash equilibrium of the meta game and* diff *be high-value uninformative. Then* $\max_{\sigma_i} u_i(\sigma_i, \sigma_{\pi_{-i}}^{\max}) \leq u_i(\pi_1, \pi_2)$ *for $i = 1, 2$.*

*Proof.* Assume for contradiction that there is some strategy $\sigma_i'$ s.t. $u_i(\sigma_i', \sigma_{\pi_{-i}}^{\max}) > u_i(\pi_1, \pi_2)$. Because diff is high-value uninformative, there exists a policy $\pi_i'$ s.t. $(\pi_i', \pi_{-i})$ resolves to a strategy profile arbitrarily close to $\sigma_i', \sigma_{\pi_{-i}}^{\max}$. Hence, $u_i(\pi_i', \pi_{-i}) > u_i(\pi_1, \pi_2)$ and so $(\pi_1, \pi_2)$ cannot be a Nash equilibrium after all. $\qquad\square$

**Lemma 23.** *Let $\Gamma$ additively decompose into $(u_{i,j} \colon A_i \to \mathbb{R})_{i,j \in \{1,2\}}$ and let* diff *be high-value uninformative. Let $(\sigma_i^{\leqslant}, \theta_i, \sigma_i^{>})_{i=1,2}$ be a Nash equilibrium. Then $u_{i,-i}(\sigma_{-i}^{\geq}) \leq u_{i,-i}(\sigma_{-i}^{\leqslant})$ for $i = 1, 2$.*

*Proof.* Follows directly from Lemma 22. $\qquad\square$

**Lemma 24.** *Let $\Gamma$ be symmetric and additively decomposable. Let* diff *be symmetric, high-value uninformative and minimized by copies. Let $(\pi_i = (\sigma_i^{\leqslant}, *, \sigma_i^{>}))_{i=1,2}$ be a Nash equilibrium of the* diff *meta game that induces strategies $\sigma_1, \sigma_2$. If $\sigma_i = \sigma_i^{>}$ for some $i$, then $(\sigma_1, \sigma_2)$ is a Nash equilibrium of $\Gamma$.*

*Proof.* With high-value uninformativeness, it follows immediately that $\sigma_{-i}$ is a best response to $\sigma_i$. The case that $\sigma_{-i} = \sigma_{-i}^{\geq}$ is trivial, so we focus on the case where $\sigma_{-i}$ gives positive probability to $\sigma_{-i}^{\leqslant}$. It then follows that $\sigma_{-i}^{\leqslant}$ is a best response to $\sigma_i$. Because the game is additively decomposable, this means that $\sigma_{-i}^{\leqslant}$ (independent of the opponent's strategy) maximizes $-i$'s utility (i.e., $\sigma_{-i} \in \arg\max_{\sigma_{-i}' \in \Delta(A_{-i})} u_{-i,-i}(\sigma_{-i}')$). So in particular, $\sigma_{-i}$ is a best response to $\sigma_i$.

It is left to show that $\sigma_i$ is a best response to $\sigma_{-i}$. We will argue that if $\sigma_i$ is not optimizing Player $i$'s utility (i.e., if $\sigma_i \notin \arg\max_{\sigma_i' \in \Delta(A_i)} u_{i,i}(\sigma_i')$), then Player $i$ could better-respond to $\pi_{-i}$ by also playing $\pi_{-i}$ instead of $\pi_i$. Let $\sigma_{-i}'$ be the strategy induced for both players by $(\pi_{-i}, \pi_{-i})$. Because diff is minimized by copies, $\sigma_{-i}'$ gives weakly more weight to $\sigma_{-i}^{\leqslant}$ than $\sigma_{-i}$. By Lemma 23, this means that $u_{i,-i}(\sigma_{-i}') \geq u_{i,-i}(\sigma_{-i})$. Second, by the assumption that $\sigma_i$ doesn't optimize $i$'s utility but $\sigma_{-i}$ and $\sigma_{-i}^{\leqslant}$ do, it follows that $u_{i,i}(\sigma_i) < u_{i,i}(\sigma_{-i}')$.

Putting it all together we obtain that

$$
\begin{aligned}
u_i(\pi_{-i}, \pi_{-i}) &= u_i(\sigma_{-i}', \sigma_{-i}') \\
&= u_{i,i}(\sigma_{-i}') + u_{i,-i}(\sigma_{-i}') \\
&> u_{i,i}(\sigma_i) + u_{i,-i}(\sigma_{-i}) \\
&= u_i(\sigma_i, \sigma_{-i}) \\
&= u_i(\pi_i, \pi_{-i}),
\end{aligned}
$$

as claimed. $\qquad\square$

### D.3 Proof of Theorem 4

**Theorem 4.** *Let $\Gamma$ be a player-symmetric, additively decomposable game. Let* diff *be symmetric, high-value uninformative, and minimized by copies. Then if $(\pi_1, \pi_2)$ is a Nash equilibrium that is not Pareto-dominated by another Nash equilibrium, we have that $V_1(\pi_1, \pi_2) = V_2(\pi_1, \pi_2)$. Hence, if there exists a Pareto-optimal Nash equilibrium, its payoffs are unique, Pareto-dominant among Nash equilibria and equal across the two players.*

For the proof we define for additively decomposable games, $u_{\Sigma,j} := u_{1,j} + u_{2,j} \colon A_j \to \mathbb{R}$. Intuitively, $u_{\Sigma,j}$ denotes the utilitarian welfare generated by Player $j$'s actions. In symmetric games, $u_{\Sigma,1} = u_{\Sigma,2}$

so that we can simply write $u_\Sigma$. For example, in the Prisoner's Dilemma $u_\Sigma \colon \mathrm{Cooperate} \mapsto G, \mathrm{Defect} \mapsto 1$.

*Proof.* We will prove that if $(\pi_i = (\sigma_i^{\lessgtr}, \theta_i, \sigma_i^{>}))$ is a Pareto-optimal equilibrium of the meta game, then both players receive the same utility. The uniqueness of the Pareto-optimal equilibrium follows immediately.

We prove this in turn by contradiction. So assume that $(\pi_1, \pi_2)$ is a Pareto-optimal equilibrium of the meta game and that the two players receive different utilities.

Assume WLOG that in $(\pi_1, \pi_2)$ Player 1 receives higher utility. Let $\sigma_1, \sigma_2$ be the strategies played in $(\pi_1, \pi_2)$. Then we distinguish two cases:

    A) Player 1 "takes" more than Player 2, i.e.,

$$u_{1,1}(\sigma_1) > u_{2,2}(\sigma_2). \tag{1}$$

    B) Player 1 does not take more but Player 2 "gives" more than Player 1, i.e.,

$$u_{1,1}(\sigma_1) \leq u_{2,2}(\sigma_2) \tag{2}$$

    and

$$u_{2,1}(\sigma_1) < u_{1,2}(\sigma_2). \tag{3}$$

It is easy to see that one of these cases must obtain.

A) We in turn distinguish two cases:

A.1) First consider the case where

$$u_\Sigma(\sigma_1^{>}) \geq u_\Sigma(\sigma_1^{\lessgtr}). \tag{4}$$

We will show that in this case $(\pi_1, \pi_2)$ cannot be a Nash equilibrium. Player 2 can better-respond by playing the policy $\tilde{\pi}_1$ that plays $\sigma_1^{>}$ and maximizes (as per the high value uninformativeness condition) Player 1's probability of playing $\sigma_1^{>}$. This can be seen as follows:

$$
\begin{aligned}
u_2(\pi_1, \pi_2) \quad &= \quad u_2(\sigma_1, \sigma_2) \\
&\underset{\text{Ineq. 1}}{<} \quad u_2(\sigma_1, \sigma_1) \\
&\underset{\text{Ineq. 4}}{\leq} \quad u_2(\sigma_1^{>}, \sigma_1^{>}) \\
&\underset{\text{Lemma 23}}{\leq} \quad u_2(\sigma_1^{\max}, \sigma_1^{>}) \\
&= \quad u_2(\tilde{\pi}_1, \pi_2)
\end{aligned}
$$

A.2) Now consider the case where

$$u_\Sigma(\sigma_1^{>}) \leq u_\Sigma(\sigma_1^{\lessgtr}). \tag{5}$$

We will show that in this case Player 2 can better respond by also playing $\pi_1$ instead of $\pi_2$, such that $(\pi_1, \pi_2)$ (again) cannot be a Nash equilibrium.

Let $\sigma_1'$ be the strategy played by both players in $(\pi_1, \pi_1)$. Note that because $\mathrm{diff}$ is minimized by copies, $\sigma_1'$ gives at least as much weight to $\sigma_1^{\lessgtr}$ as $\sigma_1$.

$$
\begin{aligned}
u_2(\pi_1, \pi_1) \quad &= \quad u_2(\sigma_1', \sigma_1') \\
&\underset{\text{Ineq. 5}}{\geq} \quad u_2(\sigma_1, \sigma_1) \\
&\underset{\text{Ineq. 1}}{>} \quad u_2(\sigma_1, \sigma_2) \\
&= \quad u_2(\pi_1, \pi_2)
\end{aligned}
$$

B) We again distinguish two cases.

B.1) First consider the case where

$$u_\Sigma(\sigma_2^>) \leq u_\Sigma(\sigma_2^\lessgtr). \tag{6}$$

We will show that in this case $(\pi_2, \pi_2)$ is also a Nash equilibrium and that $(\pi_2, \pi_2)$ Pareto-dominates $(\pi_1, \pi_2)$.

First, we show Pareto dominance. Let $\sigma_2'$ be the strategy played by both players in $(\pi_2, \pi_2)$. Because diff is minimized by copies, $\sigma_2'$ gives at least as much weight to $\sigma_2^\lessgtr$ as $\sigma_2$. Then for Player 1 we have that

$$
\begin{aligned}
u_1(\pi_1, \pi_2) = u_1(\sigma_1, \sigma_2) &\underset{\text{Ineq. 2}}{\leq} u_1(\sigma_2, \sigma_2) \\
&\underset{\text{Ineq. 6}}{\leq} u_1(\sigma_2', \sigma_2') = u_1(\pi_2, \pi_2).
\end{aligned}
\tag{7}
$$

Player 2's utility is strictly higher in $(\pi_2, \pi_2)$, which we can see as follows:

$$
\begin{aligned}
u_2(\pi_1, \pi_2) &= u_2(\sigma_1, \sigma_2) \\
&\underset{\text{Ineq. 3}}{<} u_2(\sigma_2, \sigma_2) \\
&\underset{\text{Ineq. 6}}{\leq} u_2(\sigma_2', \sigma_2') \\
&= u_2(\pi_2, \pi_2).
\end{aligned}
$$

It is left to show that $(\pi_2, \pi_2)$ is a Nash equilibrium. By assumption, $\pi_1$ is a best response to $\pi_2$. Line 7 therefore implies that $\pi_2$ is also a best response to $\pi_2$. Because of symmetry, this is true for both players. We conclude that $(\pi_2, \pi_2)$ is a Nash equilibrium.

B.2) Let

$$u_\Sigma(\sigma_2^>) > u_\Sigma(\sigma_2^\lessgtr). \tag{8}$$

We now must make one more distinction. Consider first the case where $\sigma_2 = \sigma_2^>$. By Lemma 24, $(\sigma_1, \sigma_2)$ must be a Nash equilibrium of $\Gamma$. The contradiction follows immediately from Proposition 5.

Now consider the case where $\sigma_2 \neq \sigma_2^>$. With Ineq. 8 it follows that

$$u_1(\sigma_2, \sigma_2) < u_1(\sigma_2^>, \sigma_2^>). \tag{9}$$

We will show that $(\pi_1, \pi_2)$ is not an equilibrium because Player 1 can better-respond by playing the policy $\tilde{\pi}_1$ that plays $\sigma_2^>$ and maximizes as per the definition of high-value uninformativeness Player 2's probability of playing $\sigma_2^>$. Then

$$
\begin{aligned}
u_1(\pi_1, \pi_2) &= u_1(\sigma_1, \sigma_2) \\
&\underset{\text{Ineq. 2}}{\leq} u_1(\sigma_2, \sigma_2) \\
&\underset{\text{Ineq. 9}}{<} u_1(\sigma_2^>, \sigma_2^>) \\
&\underset{\text{Lemma 23}}{\leq} u_1(\sigma_2^>, \sigma_2^{\max}) \\
&= u_1(\tilde{\pi}_1, \pi_2).
\end{aligned}
$$

$\square$

# E   Theoretical analysis beyond threshold policies

We here analyze a meta game in which players can not only submit threshold policies but continuous functions. The goal is to show an equilibrium based on linear functions, similar to the equilibria found by CCDR pretraining.

A policy now is a function $\pi\colon \mathbb{R}_{\geq 0} \to [0, 1]$, where $\pi(x)$ denotes $\pi$'s probability of cooperation. For differences $x_1, x_2$, the payoff of Player $i$ is given by

$$(1 - \pi_i(x_i)) + G \cdot \pi_{-i}(x_{-i}).$$

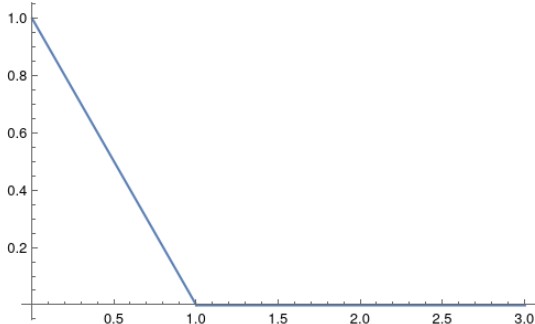

Figure 6: Visualization the continuous policy in Proposition 25 for $a = 1$. The probability of cooperation is plotted against the perceived difference to the opponent.

It is left to specify the difference function. Let $\epsilon > 0$ and $K > \epsilon$. Then define the function difference

$$d(\pi_1, \pi_2) = \frac{1}{K} \int_0^K |\pi_1(x) - \pi_2(x)| dx$$

Further, define the probabilistic difference mapping $\mathrm{diff}(\pi_1, \pi_2) = d(\pi_1, \pi_2) + Z_i$, where $Z_i$ is drawn uniformly from $[0, \epsilon]$. As the set of policies for each player consider the set of integrable functions.

**Proposition 25.** *Let* $\pi(x) = \max(1 - x/a, 0)$ *for some* $a > \epsilon$. *Then* $(\pi, \pi)$ *is a Nash equilibrium if and only if* $1 + aK/\epsilon \leq G$.

Note that $\pi$ decreases cooperation linearly in $x$ down to $0$ (which is hit at $x = a$). This function is shown in Figure 6 for $a = 1$. Note that our policies look roughly like this after Step 2.

*Proof.* Consider $(\pi, \pi)$ and imagine that, WLOG, Player 1 moves away from $\pi$ by $\Delta$, i.e., deviates to play some $\pi'$ s.t. $d(\pi, \pi') = \Delta$. It is easy to see that it is enough to consider small deviations. Specifically, we assume $\Delta \leq a - \epsilon$. First, if the difference between the policies increases by $\Delta$, what happens to Player 2's expected amount of cooperation? It is easy to see that this decreases by $\Delta/a$.

Next, we need to ask: by increasing the difference by $\Delta$, how much can Player 1 increase her probability of defection? We need to consider two effects. First, if the difference increases by $\Delta$, then automatically Player 1 defects more by the same effect as Player 2. So this gives Player 1 an extra $\Delta/a$ probability of defection. Moreover, Player 1 can decrease the probability of cooperation on the relevant interval $[\Delta, \Delta + \epsilon]$. This decreases the probability of cooperation by (at most) $K\Delta/\epsilon$.

Taking stock, Player 1 can increase her probability of defecting by $\Delta/a + K\Delta/\epsilon$ at the cost of Player 2 increasing her probability of defecting by $\Delta/a$. This is good for Player 1 if and only if $1 + aK/\epsilon > G$. $\qquad\square$

## F  Details on our experiments

### F.1  Software

We used `pytorch` for implementing CCDR and ABR and `functorch` for implementing LOLA. We used floats with double precision (by running `torch.set_default_dtype(torch.float64)`), because preliminary experiments had shown numerical issues as ABR converged. We used Weights and Biases (`wandb.ai`) for tracking.

### F.2  Game and meta game

We here give some details on the game and diff meta game we consider throughout our experiments.

**Constructing** $f_D, f_C$  In our experiments $f_D, f_C$ have input dimension 10 and output dimension 3. (Thus, including one dimension for the similarity value, our policies have input dimension 11.) First we generate $\mathbf{s}_{C,i}$ and $\mathbf{s}_{D,i}$ for $i = 1, 2, 3$ from $\{0, 1\}^{10}$ uniformly at random. Then we define

$$f_C(\mathbf{x}) = (\sin(\mathbf{s}_{C,i} \cdot \mathbf{x}))_{i=1,2,3}.$$

We define $f_D$ analogously based on $\mathbf{s}_D$.

We chose this function because it is very simple to understand and implement and at the same time requires using larger nets. The only other approach we tried in preliminary experiments is to generate $f_C, f_D$ by randomly generating neural nets. The problem is that large randomly generated fully connected neural nets are close to constant functions.

**Constructing $\mu$**    Recall that for any pair of actions $f_1, f_2$, the payoffs of the HDPD are given by $u_i(f_1, f_2) = -\mathbb{E}_{\mathbf{x} \sim \mu}[d(f_i(\mathbf{x}), f_D(\mathbf{x})) + Gd(f_i(\mathbf{x}), f_C(\mathbf{x}))]$, where $d$ is the Euclidean distance and $\mu$ is some measure of $\mathbb{R}^{10}$. Thus, to construct a specific instance of the HDPD we need to also construct $\mu$. We do this by generating 50 vectors uniformly from $[0, 1]^{10}$ and then taking the uniform distribution over these 50 vectors.

**Constructing    the    noisy    diff    mapping**  Recall    that    $\mathrm{diff}_i(\pi_1, \pi_2)$    $=$ $\mathbb{E}_{(y,\mathbf{x}) \sim \nu}[d(\pi_1(y, \mathbf{x}), \pi_2(y, \mathbf{x}))] + Z_i$, where $\nu$ is some probability distribution over $\mathbb{R}^{n+1}$ and $Z_i$ is some real-valued noise. We need to specify $\nu$ and the distribution $Z_i$. For $\nu$ we first generate 50 reals from $[0, 0.1]$ uniformly at random as our test diffs. We increment each of these by a random draw from the underlying noise distribution, i.e., by a number drawn uniformly at random from $[0, 0.1]$. We then define $\nu$ to be the uniform distribution over 50 values that result from pairing the support of $\mu$ with these 50 values.

For the noise we generate for each player 50 values uniformly from $[0, 0.1]$ and then use the uniform distribution over these 50 points.

Note that by using the uniform distribution over a finite support, we can compute expected utilities in the meta game exactly.

### F.3    Neural net policies

Throughout our experiments, our policies $\pi_{\boldsymbol{\theta}}$ are represented by neural networks with three fully connected hidden layers of dimensions 100, 50 and 50 with biases and LeakyReLU activation functions. Thus, these networks overall have $11 + 100 + 50 + 50 + 3 + 11 \cdot 100 + 100 \cdot 50 + 50 \cdot 50 + 50 \cdot 3 = 8964$ parameters.

### F.4    Methods as used in our experiments

**CCDR pretraining**    We here describe in more detail the CCDR pretraining step in our results. Recall that CCDR pretraining consists in maximizing $V^d(\pi_{\theta_i}, \pi_{\theta_i}) + V^d(\pi_{\theta_i}, \pi'_{\theta_{-i}})$ for randomly generated opponents $\pi'_{\theta_{-i}}$. Call $-V^d(\pi_{\theta_i}, \pi_{\theta_i}) + V^d(\pi_{\theta_i}, \pi'_{\theta_{-i}})$ the CCDR loss.

In our experiments, we maximized this by running Adam for 100 steps. In each step, the CCDR loss is calculated by averaging over 100 randomly generated opponents. The learning rate is 0.02.

**Alternating best response training**    We ran alternating best response training for $T = 1000$ turns. We need to specify how we updated $\theta_i$ to maximize $V(\theta_i, \theta_{-i})$ (holding $\theta_{-i}$ fixed) in each turn. For this we run gradient descent for $T' = 1000$ steps. However, we only take gradient steps that are successful, i.e., that in fact reduce loss. The learning rate $\gamma'$ is sampled uniformly from $[0, \gamma = 0.00003]$ in each step. Note that by randomly sampling the learning rate, the algorithms avoids getting stuck when a gradient step is unsuccessful. We summarize this in Algorithm 1.

### F.5    Convergence to spurious stable points?

In theory, alternating best response learning could converge to a "very local Nash equilibrium", i.e., a pair of models that are best responses only within a very small neighborhood of these models. One might also worry about convergence to other stable points (Mazumdar et al., 2020).

However, as far as we can tell, the limit points of our learning procedure are not spurious. To confirm this, we performed the following test (implemented by the function `best_response_test` of our code). For a given pair of models, we pick either model and perturb each of its parameters a little. We then see whether the perturbed model is a better response to the opponent model than the original model. In an (approximate) local Nash equilibrium, this should almost never be the case.

In all but one of our successful runs (the ones converging to partial cooperation), none of 10,000 random perturbations of each of the models after alternating best response training led to a better response to the opponent model. In one run, three perturbations of one of the models decreased loss.

---

**Algorithm 1** Alternating best response learning

---

**Input:** Initial model parameters $\theta_1, \theta_2, T, T' \in \mathbb{N}$, learning rate $\gamma$
**Output:** New model parameters $\theta_1', \theta_2'$
**for** $t = 1, ..., T$ **do**
    **for** $i = 1, 2$ **do**
        **for** $t' = 1, ..., T'$ **do**
            $\gamma' \sim \mathrm{Uniform}([0, \gamma])$
            $\theta_i'' \leftarrow \theta_i' + \gamma' \nabla_{\theta_i'} V(\theta_i', \theta_{-i}')$
            **if** $V(\theta_i'', \theta_{-i}') \geq V(\theta_i', \theta_{-i}')$ **then**
                $\theta_i' \leftarrow \theta_i''$
            **end if**
        **end for**
    **end for**
**end for**
**return** $\theta_1', \theta_2'$

---

When applying the test after CCDR pretraining but before alternating best response training, close to half of random perturbations improve utility.

### F.6 Compute costs

We here provide details on how computationally costly our experiments are. To do so, we ran the experiment for a single random seed on an AMD Ryzen 7 PRO 4750U, a laptop CPU launched in 2020. (Note that we used the CPU not GPU (CUDA).) The CCDR pretraining took about 30 seconds per model. The ABR phase took about 3h. (Note that we ran most of our experiments as reported in the paper via remote computing on different hardware.)

## G Experiments with LOLA

We have tried to learn SBC using Foerster et al.'s (2018) Learning with Opponent-Learning Awareness (LOLA). We here specifically report on an experiment in which we tried a broad range of parameters. The results are in line with prior exploratory experiments.

LOLA sometimes succeeds in learning SBC (without CCDR pretraining). It finds similar policies as CCDR pretraining. Some of the SBC models found by LOLA remain cooperative throughout a subsequent ABR phase.

Unfortunately, none of our LOLA results are nearly as robust as the CCDR results reported in the main text and in Appendix F. Specifically, they are not even robust to changing the random seed. One reason for this is that LOLA is capable of "unlearning" LOLA-learned SBC.

### G.1 Experimental setup

We ran Foerster et al.'s (2018) Learning with Opponent-Learning Awareness (LOLA). We tried every combination setting the learning rate and lookahead parameter in LOLA to values in 0.0003, 0.001, 0.003, 0.01, 0.03, 0.1, 0.3, 1. For each combination we tested both *exact* and Taylor LOLA as described by Willi et al. (2022, Sect. 3.2). We tested each of these $8 \cdot 8 \cdot 2 = 128$ combinations with 5 different random seeds. In each of these experiments we initialized the nets randomly and then ran LOLA for both agents for 30,000 steps. We then ran ABR training as described in Appendix F.4 for at most 500 steps, to test whether the resulting policies are in equilibrium. To save compute, we generally aborted ABR when the loss hit the Defect–Defect utility of 5 or when it was clear that ABR learning had converged.

We then labeled each individual run as a "weak LOLA success" if the loss after the LOLA phase was smaller for both players than the loss of mutual defection and as a LOLA-ABR success if the loss after the ABR phase was smaller for both players than the loss of mutual defection.

For each parameter configuration in which at least 3 out of 5 runs were a weak LOLA success we ran another 20 runs (with different random seeds) without the ABR phase to determine how robust the success is. Similarly, for each parameter configuration in which at least 3 out of 5 runs were a LOLA-ABR success we ran another 20 runs *with* the ABR phase.

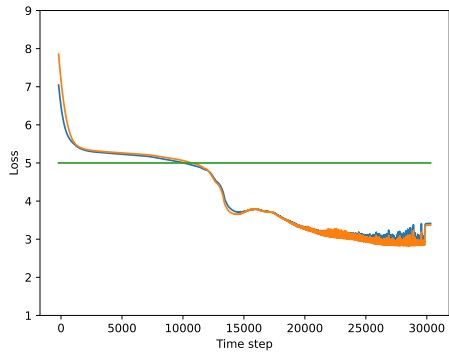

Figure 7: Loss curve of a relatively successful run between two LOLA agents.

## G.2 Qualitative analysis

We start with a qualitative discussion of our results, because the experiments with LOLA exhibit a much wider range of complex phenomena than the CCDR-based results.

### G.2.1 A few clear-cut successful runs

Out of all the runs in our experiment, a few are as clear-cut successes as most of the CCDR runs. Figure 7 shows the learning curve of one of these runs across both the LOLA and at the end the ABR phase. It uses a lookahead of $0.001$, learning rate of $0.001$ and exact LOLA. The players learn to cooperate with each other in the LOLA phase. When the switch to ABR is made after 30k steps of LOLA, cooperation deteriorates slightly, but ultimately the models converge in the ABR phase to an outcome with a loss well below the loss of mutual defection. This shows that LOLA is to some extent capable of solving the given problem.

Unfortunately, both in the present experiment and in preliminary experiments we have found that these results are not even robust to changing the random seed. For example, the four other runs with the same parameters as the run from Figure 7 (but different random seeds) all failed to produce significant positive results. (One resulted in marginal cooperation at the end of the LOLA phase (losses 4.986 and 4.928) that disappeared immediately when switching to ABR learning. Another partially cooperated for some of the LOLA phase but failed to retain cooperation even until the end of the LOLA phase.)

### G.2.2 What models LOLA learns

Generally, when LOLA succeeds, it learns similar models as CCDR pretraining. The main difference is that for high diff values, LOLA models typically do not exactly defect. (Of course, it also does not cooperate. Instead if performs an action far from both $f_C$ and $f_D$.) This is to be expected, since these agents are not trained on randomly generated opponents. An example model after 30,000 steps of LOLA is shown in Figure 8.

When LOLA does not succeeed, the learned models vary greatly. Some are clearly set up to impose some incentives – others just constantly defect.

### G.2.3 Instability in the LOLA phase

The learning in the LOLA phase is often highly unstable. That is, the during the learning phase, the change in losses in a single learning step is often very large. This occurs even in successful runs. As an example, consider the learning curve in Figure 9 (lookahead $0.003$, learning rate $0.01$, exact LOLA). While it is unclear whether cooperation would have survived further ABR learning, this run is relatively successful. However, the utilities vary by large amounts during the LOLA phase. In fact for parts of the LOLA learning phase, both players' losses are much higher than the loss of mutual defection. Most successful runs with LOLA look like this. Of course, successes based on such runs cannot be very robust: if we had ceased LOLA learning somewhat earlier, the run would not have succeeded. It is unclear what would happen if the run had run for more than 30k LOLA steps.

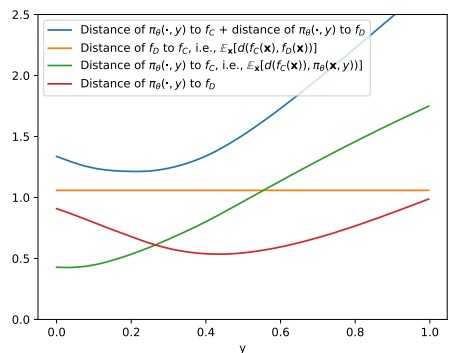

Figure 8: An example of a model found in a successful LOLA run.

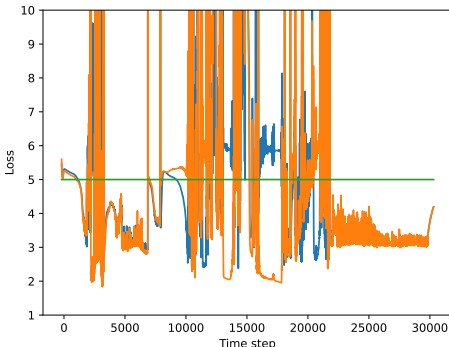

Figure 9: An example of the loss curve from a highly unstable LOLA run.

### G.2.4 Cooperation in the LOLA phase versus cooperation in the ABR phase

While CCDR is explicitly a *pre*training method, LOLA can also be used as a standalone learning procedure. So for each run, we can ask both whether the players cooperate, say, at the end of the LOLA phase and whether the players cooperate at the end of the ABR phase. It turns out that all four combinations of answers are possible! In particular, there are runs in which LOLA versus LOLA fails to converge to a pair of policies that give both players a higher utility than mutual defection; but in which ABR then converges to an outcome that is marginally better for both players than mutual defection.

### G.3 Quantitative analysis

Bearing in mind the different qualitative phenomena, we here give a more quantitative analysis.

Table 5 lists the parameters in which at least three out of five runs exhibited weak LOLA success as defined earlier. They also summarize the result of 20 further runs with these parameters. Specifically, we categorized runs as *stable successes* if the loss was below the loss of mutual defection for both players for the final 5,000 LOLA steps. We categorized them as *unstable successes* if the loss was below the loss of mutual defection for both players for the final 50 but not the final 5,000 LOLA steps. The fourth and fifth columns of Table 5 show the numbers of these different kinds of successes out of 20 runs. The sixth colum shows the average loss across both players and all successful runs. The final column shows the standard deviation across the losses of the successful runs (also across players).

Table 6 lists the set of parameter configurations that achieved a LOLA–ABR success in at least three out of five runs. They also summarize the result of 20 further runs with each of these parameter configurations. Specifically, we categorized runs again as stable successes if the loss was below the loss of mutual defection for both players for the final 5,000 LOLA steps *and* at the end of the ABR phase. We labeled them as unstable successes if the loss was below the loss of mutual defection for both players at the end of the ABR phase but *not* for the final 5,000 LOLA steps. Again, the

| LOLA LA | LOLA LR | Taylor LOLA | Stable Successes | Unstable Successes | average success loss | SD |
|---|---|---|---|---|---|---|
| 0.003 | 0.0003 | false | 5 | 1 | 4.518 | 0.2857 |
| 0.001 | 0.01 | false | 1 | 0 | 4.666 | 0.002601 |
| 0.001 | 0.003 | false | 5 | 7 | 3.579 | 0.6096 |
| 0.0003 | 0.003 | true | 3 | 7 | 4.299 | 0.1482 |
| 0.0003 | 0.003 | false | 4 | 3 | 4.506 | 0.07975 |
| 0.0003 | 0.001 | true | 4 | 1 | 4.344 | 0.2466 |

Table 5: Parameters for which at least three of five runs resulted in partial cooperation at the end of the LOLA phase along with results of further 20 runs per parameter configuration..

| LOLA LA | LOLA LR | Taylor LOLA | Stable Successes | Unstable Successes | average success loss | SD |
|---|---|---|---|---|---|---|
| 0.0003 | 0.003 | true | 1 | 12 | 4.279 | 0.2830 |
| 0.0003 | 0.003 | false | 3 | 9 | 4.344 | 0.2566 |

Table 6: Parameters for which at least three of five runs resulted in partial cooperation at the end of the ABR phase along with results of further 20 runs per parameter configuration.

fourth and fifth columns of Table 6 show the numbers of these different kinds of successes out of 20 runs. The sixth column shows the average loss across both players and all successful runs. The final column shows the standard deviation across the losses of the successful runs (also across players).

From these results we conclude that LOLA can succeed under various parameter configurations, but none of the parameter configurations succeed nearly as reliably as CCDR pretraining.

### G.4 Compute costs

We again provide details on how computationally costly our experiments are. To do so, we ran the experiment for a single random seed on an AMD Ryzen 7 PRO 4750U. The LOLA phase took about 20 minutes. The ABR phase took about 1h and 30min. Note again we ran most of our experiments as reported in the paper via remote computing on different hardware. Note also that we ran ABR for half as many steps compared to the experiments for the main text (500 instead of 1000). So, the cost per ABR step is roughly the same between the two experiments.

## H   Distantly related work

### H.1   Learning in Newcomb-like decision problems

There is some existing work on learning in Newcomb-like environments that therefore also applies to the Prisoner's Dilemma against a copy. Whether cooperation against a copy is learned generally depends on the learning scheme. Bell et al. (2021) show that $Q$-learning with a softmax policy learns to defect. Regret minimization also learns to defect. Other learning schemes do converge to cooperating against exact copies (Albert & Heiner, 2001; Mayer et al., 2016; Oesterheld, 2019a; Oesterheld et al., 2023). All schemes in prior work differ from the present setup, however, and to our knowledge none offer a model of cooperation between similar but non-equal agents.

### H.2   Cooperation via iterated play, reputation, etc.

Perhaps the best-known way to achieve cooperation in the Prisoner's Dilemma is to play the Prisoner's Dilemma repeatedly (e.g., Axelrod 1984; Osborne 2004, Ch. 14, 15). Clearly, the underlying mechanism (repeated play) is very different from the mechanism underlying SBC, in which the game is played one-shot. That said, our folk theorem (Theorem 3) is similar to the well-known folk theorem for repeated games. (As noted in Section 7, the folk theorem for program equilibrium is also similar.) A number of variants of iterated play have been considered to study, for example, reputation effects, effects of allowing players to choose with whom to play based on reputation, and so on (e.g., Nowak & Sigmund 1998).

While cooperation via repeated play is very different from similarity-based cooperation as studied in this paper, some variants of the former are easy to confuse with the latter. As a simplistic example, consider a variant of the infinitely repeated Prisoner's Dilemma. Typically, it is imagined that players observe the entire history of past play. But now imagine that instead the players in each round only observe a single bit: 0 if they have taken the same action in each round so far and 1 otherwise.

Then it is a (subgame-perfect) Nash equilibrium for both players to follow the following strategy: cooperate upon observing 0 and defect upon observing 1. This is somewhat similar to similarity-based cooperation, but the underlying mechanism is still the one of repeated play: the reason it is rational for each of the players to cooperate is that if they defect, their opponent will then defect in all subsequent rounds. In fact, the strategies played are equivalent to the grim trigger strategies for the iterated Prisoner's Dilemma.

As another example of cooperation via repeated play that can be confused with SBC, consider assortative matchmaking as proposed by Wang et al. (2018, Sect. 2.4). Imagine that agents in a large population are repeatedly paired up to play a Prisoner's Dilemma. In each round, each agent is matched up with a *new* opponent who is similarly cooperative as they are. Again, similarity plays a role, but the underlying mechanism is very different from the ones studied in our paper. For one, similarity is used by the environment (the matching procedure) not the agent itself. Second, the reason why cooperation is rational is again its impact on rewards in *subsequent* rounds. For instance, imagine that for some reason an agent has, e.g., by accident, defected in the first few rounds and is now matched with uncooperative agents. Then it may be rational for the agent to cooperate in order to be matched with more cooperative agents in the future, even if it knows that it will receive the sucker's payoff in the current round.

### H.3 Tag-based cooperation

In the literature the evolution of cooperation and in particular population dynamics, researchers have studied so-called *tag-based cooperation* Riolo et al. (2001); Cruciani et al. (2017); Traulsen & Claussen (2004); Traulsen (2008), which has sometimes also been referred to as similarity-based cooperation. The underlying mechanism is very different from similarity-based cooperation as studied in the present paper, however.

In models of tag-based cooperation, an agent is defined not only by a policy but also by a *tag* that has nothing to do with the policy. When choosing whether to cooperate with each other, agents can see each other's tags. Thus, the policies can choose based on the other agent's tag. For example, one policy could be to cooperate if and only if the two agents have the same tag. Notice that in tag-based cooperation, only similarity of tags is considered. In contrast, our paper feeds the similarity of the policies themselves as input to the policy.

Because the tag is not tied to the policy, tags are effectively cheap talk that have no impact on the Nash equilibria of the game. For example, consider the following meta game on the Prisoner's dilemma. Each player $i$ submits a tag $\tau_i \in \{1, ..., 100\}$ and a policy $\pi_i \colon \{1, ..., 100\} \rightsquigarrow \{C, D\}$ that stochastically maps opponent tags onto actions. Then actions are determined by $a_i = \pi_i(\tau_{-i})$ for $i = 1, 2$. Each player $i$ receives the utility $u_i(a_1, a_2)$. It is easy to see that in all Nash equilibria of this game, both players submit a policy that always defects. So tag-based cooperation does not help with achieving cooperative equilibrium in a two-player Nash equilibrium. In contrast, similarity-based cooperation as studied in this paper *does* allow for cooperative equilibria in two-player games, as the main text has shown.

So why might one study tag-based cooperation? Perhaps one would expect that any population of evolving agents would converge to defecting in the above tag-based Prisoner's Dilemma. Interestingly, it turns out that this is not the case! As computational experiments conducted in the above works show, evolving populations sometimes maintain some level of cooperation when they can observe each others' tags. Because cooperation is not an equilibrium, cooperation is maintained dynamically, i.e., by a population whose distribution of tags and policies continually changed. Roughly, the reason seems to be the following. By mere chance (from mutation) there will be tags whose individuals are more cooperative toward each other than the individuals associated with other tags. The cooperative tags and their associated cooperative policies will therefore become more prominent in the population. Of course, this cannot be stable: once an agent is discovered that has the cooperative tag but defects, this type of agent takes over within the cooperative population. But in the meantime a new cooperative tag may have emerged. And so forth. This mechanism for achieving cooperation seems to have no parallel at all in our model of similarity-based cooperation.[1]

---

[1]See also Nowak & May (1992) for another line of work that is even more different from the present paper, but in which unstable cooperation survives dynamically by similar mechanisms.

## H.4 The green beard effect

The *green beard effect* (Hamilton 1964; Dawkins 1976, Ch. 6; Gardner & West 2010) refers to the idea that there could be a gene $\mathcal{G}$ that causes individuals to be altruistic to people who also have the gene $\mathcal{G}$ *without* relying on kinship to identify other people with the gene $\mathcal{G}$. In particular, imagine a gene $\mathcal{G}$ with the following three properties:

1. Gene $\mathcal{G}$ causes people with the gene to have some perceptible trait. For concreteness let the trait consist in growing a green beard.
2. Gene $\mathcal{G}$ causes people to be altruistic toward people who have a green beard, e.g., to cooperate with them in a one-shot Prisoner's Dilemma.
3. People without gene $\mathcal{G}$ cannot develop a green beard.

Then such a gene could spread and be dominant in a population.

Hamilton and Dawkins discuss the green beard effect as a theoretical idea. To what extent the green beard effect is a real biological phenomenon is subject to debate. Some candidates were pointed out by Keller & Ross (1998), Queller et al. (2003), Smukalla et al. (2008) (cf. Sinervo et al., 2006).

On first sight, similarity-based cooperation as studied in this paper and the green beard effect might seem similar. For example, if you observe a population of individuals, some of whom have the green beard and some of whom do not, you will see cooperation between agents that are similar (in that they both have the green beard). A connection between similarity-based cooperation and the green beard effect has been noted before by Howard (1988), Štěpán Veselý (2011), and Martens (2019).

On second sight, however, we think that similarity-based cooperation as studied in our model is very different from the the green beard effect as an evolutionary phenomenon. To understand why, we take the gene-centered view of evolution (e.g. Dawkins, 1976): we view evolution as a process that produces *genes* that take measures to spread themselves (as opposed to a process that creates organism with particular properties). Now imagine that two individuals sharing the gene $\mathcal{G}$ play a Prisoner's Dilemma where they can either take 1 unit of a resource for themselves or give $G > 1$ units of the resource to the other player. Then from the perspective of trying to spread $\mathcal{G}$, this is not a Prisoner's Dilemma at all! (This is assuming that both players can make similarly good use of the resources.) From the gene's perspective, it is a fully cooperative game wherein cooperation simply dominates defecting. This is the essence of how the green-beard effect works. Our model of SBC has no analogous mechanism or perspective.

As a result, the green beard effect allows for a very different set of outcomes than our folk theorem (Theorem 3). For example, imagine that organisms 1 and 2 both share $\mathcal{G}$, that organism 1 can either take 1 unit of a resource for itself or give $G > 1$ units of the resource to organism 2, and that organism faces no choice that concerns organism 1. Then $\mathcal{G}$ would still have the organism give, because doing so increases the spread of the gene. Meanwhile, the unique outcome allowed by our folk theorem has organism 1 *not* give. In other games, our folk theorem allows for many different equilibria of the meta game. In contrast, in a game played by two organisms who share $\mathcal{G}$, there will typically be a unique outcome that best spreads $\mathcal{G}$.

## H.5 Studies of human psychology on similarity-based cooperation

In this section, we review empirical and theoretical work in psychology related to the phenomenon of similarity-based cooperation.

Studies have shown that humans often cooperate in the one-shot Prisoner's Dilemma, contrary to what standard game theory recommends. Many explanations have been given for this phenomenon. A few authors have proposed explanations based on Newcomb's problem as discussed in Section 7. That is, they have proposed that people cooperate in the Prisoner's Dilemma because cooperating gives them evidence that their opponent will defect (Krueger & Acevedo, 2005; Krueger et al., 2012). To test this hypothesis, a few studies have tried to measure the correlation in subjects' choice in Newcomb's problem and the Prisoner's Dilemma. The results have been mixed. Goldberg et al. (2005, Sect. 4.3), Tversky & Shafir (1992) have found strong correlations, while Toplak & Stanovich (2002) and Goldberg et al. (2005, Sect. 4.2) have failed to find such correlations. Another prediction of the Newcomb's problem/SBC explanation for human cooperation in the one-shot Prisoner's Dilemma is that individuals are more likely to cooperate with similar than with dissimilar individuals. Indeed, a number of studies have found this to be the case (e.g. Acevedo & Krueger, 2005; Fischer, 2009). (Note that some other theories have been proposed for this phenomenon as well (e.g. Aksoy, 2015).)

Another related idea in social psychology is *homophily*, which refers to the observed tendency of humans to be more likely to engage in interactions with similar individuals. (Homophily cannot always be cleanly differentiated from the tendency to be more altruistic/cooperative toward similar individuals as discussed in the previous paragraph.) Homophily could also be explained by SBC, because SBC allows for better outcomes (e.g., mutual cooperation over mutual defection) when playing against similar individuals. We are not aware of any discussion connecting homophily with Newcomb's problem or the idea that it could be rational to cooperate in a Prisoner's Dilemma against a copy. The theory of the evolution of homophily seems to be that individuals with a similar cultural background can more effectively communicate (and coordinate) (Fu et al., 2012). This is different from SBC as studied in the present paper.

