# OpenReview forum: "Similarity-based cooperative equilibrium"
_NeurIPS.cc/2023/Conference — NeurIPS 2023 poster_

### Official Review · Reviewer_ty7R · 2023-06-11

**Soundness:** 4 excellent
**Presentation:** 4 excellent
**Contribution:** 4 excellent
**Rating:** 8
**Confidence:** 4

**Summary:**

The paper proposes similarity-based cooperative equilibrium, which extends program equilibrium to a setting of partial transparency. This modification is more practical (as full transparency is much less realistic and is hard to work with) and has useful theoretical properties (e.g., the folk theorem states that by choosing an appropriate similarity measure, one can implement the same cooperative equilibria as if given full transparency). A simple (deep) RL algorithm is proposed to find the proposed equilibrium, which is tested in a newly proposed high-dimensional variant of PD.

**Strengths:**

- The paper builds on existing concepts like program equilibrium to propose a logical extension that is more practical and relevant for AI.
- The paper presents a strong theory followed by limited but representative experiments.
- The paper is well-written and well-structured. I appreciate the examples in the introduction and the explanations throughout the paper.
- HDPD is a cool independent contribution.
- Experiments with a relevant baseline, LOLA, are provided.

**Weaknesses:**

The only potential reason to not accept this paper is if other reviewers find problems with the theory (e.g., proofs), as I have only glanced over it. Otherwise, I do not see reasons to not accept this paper.

**Questions:**

- There is some very recent work about cooperative equilibria + MARL that could be discussed/mentioned in the camera ready. https://dl.acm.org/doi/10.5555/3545946.3598670 https://arxiv.org/abs/2305.06807 https://dl.acm.org/doi/10.5555/3545946.3598618

**Limitations:**

- The theoretical analysis is restricted to a certain class of diff-based policies (lines 161-163).
- As mentioned in section 6.4, the proposed empirical method may have limited applicability in more complex games.

---

> ### Author Rebuttal · Authors · 2023-08-10
>
> We thank Reviewer ty7R for their efforts in evaluating our manuscript! We are glad that the reviewer found the paper so interesting.
>
> >There is some very recent work about cooperative equilibria + MARL that could be discussed/mentioned in the camera ready. https://dl.acm.org/doi/10.5555/3545946.3598670 https://arxiv.org/abs/2305.06807 https://dl.acm.org/doi/10.5555/3545946.3598618
>
> We will make sure to include the referenced articles in an updated version of our manuscript. These works show how mediators and contracts can help achieve cooperative equilibrium in an ML context. We currently only reference more theoretical papers on these topics (e.g., Monderer and Tennenholtz 2009). These papers will therefore be a valuable addition to our manuscript. Thanks for bringing these works to our attention!

---

> > ### Comment · Reviewer_ty7R · 2023-08-10
> > **Rebuttal Acknowledgement**
> >
> > My pleasure!

---

### Official Review · Reviewer_xnVF · 2023-07-03

**Soundness:** 3 good
**Presentation:** 3 good
**Contribution:** 2 fair
**Rating:** 4
**Confidence:** 3

**Summary:**

The paper presents 2-player “difference meta games,” which augment standard one-shot games with similarity information about the other player.  The paper shows that such games have Pareto optimal cooperative equilibria that cause (seemingly naive) “ML algorithms” to cooperate rather than defect.  Results are established primarily theoretically.  Additionally, the CCDR algorithm is presented that cooperates in self play in a high-dimensional one-shot prisoner’s dilemma (formulated as a difference meta game).


**Strengths:**

- The paper presents an interesting: difference meta games.  These games are simpler than program meta games, and are in important ways more realistic.

- The paper has a high-level or rigor and is, for the most part, easy to read and understand.


**Weaknesses:**

- While difference meta games overcome the issue of full transparency, coming up with a reliable similarity metric (that can’t be duped) seems unrealistic.  At the very least, there seems to be very limited situations in which a difference meta game could be used.

- The CCDR algorithm seems to rely on the idea of their being “cooperate” and “defect” actions.  This seems problematic to me.  First, for an arbitrary game, I didn’t see in the paper where the high-level actions of “cooperate” and “defect” were defined.  Thus, it seems unclear what CCDR actually does.  (Perhaps I missed something obvious, for which I apologize).  Second, It seems that many games would not have such high-level actions (e.g., chicken or battle of the sexes).  From my perspective, this further limitation seems rather confounding.


**Questions:**

Practically speaking, how often in the world would an AI agent actually encounter a scenario in which a different meta game could be used?

**Limitations:**

The paper points out some limitations as “future work,” including that the best response can perform poorly against random agents and that the CCDR approach  However, it seems there are many more issues that could be made more explicit in the conclusions of the paper.  First, the CCDR algorithm appears to rely on the fact that there are “cooperative” and “defect” higher-level actions.  However, many games don’t have such actions.  Second, the approach is also limited to two players.

---

> ### Author Rebuttal · Authors · 2023-08-10
>
> We thank Reviewer xnVF for their thoughtful comments!
>
> We address the question about realism and whether the similarity metric can be duped in the overall response, because Reviewer EVhh asks a similar question.
>
> >the CCDR algorithm appears to rely on the fact that there are “cooperative” and “defect” higher-level actions. However, many games don’t have such actions.
>
> We believe this is a misunderstanding. CCDR does not assume that there is an action labeled "Cooperate" or "Defect". As described in Sect. 6.1, the core idea of CCDR is, roughly, to minimize the following loss function in pretraining: the loss obtained against an exact copy plus the loss against a randomly generated opponent. To maximize the first summand one generally has to play the highest-payoff symmetric strategy profile in the base game when observing a diff value of 0. In the Prisoner’s Dilemma, the best symmetric strategy profile is (Cooperate, Cooperate). So, that’s where the CC (“Cooperate against Copies”) comes from. However, in other games this CC summand of the CCDR loss is still well defined and maximized by what we would consider to be cooperative strategy profiles. For example, in the Game of Chicken, one would learn to play Swerve against a copy (depending on the specifics of the payoffs) and in Stag Hunt one would learn to play Stag against a copy. To minimize the second summand of the CCDR loss, one generally has to learn something like the following: If I observe that the opponent is very different from me (as different as a randomly generated opponent), then play a strategy for the base game that is good against a randomly generated opponent strategy. In the Prisoner’s Dilemma, this specifically means learning to defect upon observing that the opponent is very different (as different as a randomly generated opponent). That’s where the DR (“Defect against Random”) comes from. DR is also well defined in other games. However, compared to CC, it’s less clear what DR implies in other games, because it depends on the distribution of opponents generated. That said, we might imagine that in Stag Hunt, minimizing DR loss would imply playing Hare at high diff values due to [risk dominance](https://en.wikipedia.org/wiki/Risk_dominance).
>
> We will make sure to make it clear in the paper that CCDR doesn’t require that we have actions labeled “cooperate” or “defect”.

---

> > ### Comment · Reviewer_xnVF · 2023-08-12
> >
> > Thanks for the clarifications and arguments.  I remain dubious about the generality and practicality of the approach.

---

### Official Review · Reviewer_EVhh · 2023-07-05

**Soundness:** 3 good
**Presentation:** 4 excellent
**Contribution:** 3 good
**Rating:** 6
**Confidence:** 4

**Summary:**

Training AI agents that will reliably cooperate, both with humans and with other AI, is one of the principal goals of AI alignment research, and the prisoner's dilemma (PD) is simple game that has long been used to study cooperation.  PD is interesting because it has only one Nash equilibrium: to defect (i.e. not cooperate), which is a somewhat paradoxical result because mutual cooperation would be better for all participants.  The usual modification to PD is the iterated prisoner's dilemma, in which each agent can observe a history of past interactions.  This history opens up new equilibria, and allows agents to learn to cooperate.

This paper proposes another potential modification to PD with cooperative equilibria, namely supplying each agent with a measure of the similarity between its policy, and the policy of other agents.  Agents can learn to cooperate with similar agents, and defect against dissimilar ones.  The paper proves some basic theorems about equilibria in such games.

The paper also proposes a variant -- high-dimensional PD (HDPD), which has a more complex notion of "cooperate" and "defect" than simple PD.  Naive training of an agent in HDPD does not result in cooperation.  However, the authors introduce a pre-training mechanism called Cooperate against Copies and Defect against Random (CCDR), in which agents alternate self-play (i.e. play against perfectly similar copies) and play against random opponents.  When using this pre-training mechanism, agents learn to cooperate.

**Strengths:**

Although it is not mentioned by the authors in the paper, I would like to note that "perceived similarity" does, in fact, appear frequently in the natural world.  Kin selection (genetic similarity) is the basis for cooperation among social insects like ants and bees, and shared language, culture, and religion have historically been the basis for cooperation among humans.  IMO, this is thus a potentially important avenue of research, which has been overlooked in the literature.

The paper is well written, and the results seem solid.  The authors have an excellent discussion of the limitations of CCDR -- in particular, after learning to cooperate during pre-training, some agents partially unlearn it during the rest of training.   Further exploration in this area could have potential applications on AI alignment research.


**Weaknesses:**


My main concern is this is mostly a paper on game theory, so I question whether NeurIPS is really the right venue for it.  The actual experiment also seems somewhat simplistic to me.  I believe that the primary application of this work relates to AI alignment, so I would have preferred to see more discussion about how these ideas could be applied to the real world.


**Questions:**


From an AI alignment perspective, can a meaningful measure of "similarity" can be derived for more complex agents, operating in more complex environments?  In more complicated cases, the only reliable information that an agent can observe about other agents is the history of past actions, in which case the notion of "similarity" becomes much the same as the iterated PD, or other reputation-based systems.

Is it possible for agents to game the system, and hack the similarity function?  This is probably relevant only for more complex agents than those considered in this paper, but is extremely import wrt. to AI alignment.

Is it possible to derive a notion of similarity that would enable otherwise dissimilar agents to cooperate?  E.g. among humans, people with very different goals and backgrounds can cooperate if both parties agree to follow the same "moral code", or "rule of law".


**Limitations:**


Although this paper has potential applications wrt. to AI alignment, the authors focus primarily on detailed mathematical analysis, and fail to discuss the larger social implications.

However, I applaud the authors for their frank and open discussion of the limitations and failures of CCDR.  IMO, that discussion is one of the most valuable parts of the paper.

---

> ### Author Rebuttal · Authors · 2023-08-10
>
> We thank EVhh for this insightful review! We address the point about realistic settings and gaming the similarity metric in the general response.
>
> >My main concern is this is mostly a paper on game theory, so I question whether NeurIPS is really the right venue for it.
>
> First, we believe our paper is a valuable and relevant contribution to the NeurIPS community since the diff game setup is specifically motivated by ML agents. We also introduce a new type of prisoner’s dilemma environment for experimental work, a simple pre-training method and experimental results, including a comparison to a canonical baseline from the multi-agent learning literature. Even if the method and experimental results are limited in scope, we believe this work will serve as a foundation for future work on the important topic of cooperation in multi-agent machine learning.
>
> Second, it has long been the case that NeurIPS accepts a significant number of game theory papers, sometimes even without any strong learning component. [Some examples from previous years: https://proceedings.neurips.cc/paper_files/paper/2022/file/aa5f5e6eb6f613ec412f1d948dfa21a5-Paper-Conference.pdf ; https://proceedings.neurips.cc/paper_files/paper/2022/file/9d823334fdccb62a544fa7643cf0615d-Paper-Conference.pdf ; https://proceedings.neurips.cc/paper/2018/file/a9a1d5317a33ae8cef33961c34144f84-Paper.pdf ; https://proceedings.neurips.cc/paper_files/paper/2021/file/09a5e2a11bea20817477e0b1dfe2cc21-Paper.pdf ; https://proceedings.neurips.cc/paper_files/paper/2022/file/cfce833814505906445f8df2f65ab548-Paper-Conference.pdf ] This shows that contributions in this area are frequently evaluated as relevant to the NeurIPS community. In addition, the [NeurIPS 2023 call for papers](https://neurips.cc/Conferences/2023/CallForPapers) explicitly includes topics such as “algorithmic game theory” and “economic aspects of machine learning”. Of course, which topics/areas are relevant to NeurIPS is not our decision to make, but for the purposes of fairness and of accepting the best papers within each area, we believe this decision should be made consistently across the conference, not ad hoc on a paper by paper basis.
>
> >Is it possible to derive a notion of similarity that would enable otherwise dissimilar agents to cooperate? E.g. among humans, people with very different goals and backgrounds can cooperate if both parties agree to follow the same "moral code", or "rule of law".
>
> This is an important question. We believe the answer is yes, both in theory and practice! As the reviewer hints, the key is to use a notion of similarity that compares only cooperativeness or “moral code” and related concepts, and ignores aspects that are irrelevant to SBC. Theoretically it’s not too difficult to show the existence of such cooperative equilibria in some simple settings. Here is a simplistic example: Imagine that the two agents play a Prisoner’s Dilemma against each other, but also engage in some other activity that does not affect the other player’s payoff. Imagine further that the diff function only compares the Prisoner’s Dilemma parts of the agent’s policy, i.e., the functions from diff to probability of cooperation. Then all results from our paper can be directly applied to show that cooperative equilibria exist. In any case, our folk theorem implies the existence of equilibria between agents that behave very differently, though there might not be any equilibria with natural diff functions.
>
> We believe that this SBC between otherwise dissimilar agents can also work in practice, because, again, for SBC only the similarity of a very specific aspect of the agents and their policies matters. That said, one should expect some difficulties in the real world, because it is often not clear whether a particular signal of being different matters or not. Further experimental work is needed to evaluate how much is lost to these difficulties.
>
> We will update our manuscript to make it clearer that only a specific kind of similarity is important. We will also add a note on this in the “Conclusion and future work” section.
>
> With regard to AI alignment, we will elaborate on the relevance of this work in the camera ready copy. We think our work is important for two reasons. First, we want future models acting as representatives of different human actors to be able to reach and enforce mutually beneficial, cooperative outcomes and avoid conflict (see the recent agenda papers by Dafoe et al. (https://arxiv.org/abs/2012.08630) and Conitzer et al. (https://doi.org/10.1609/aaai.v37i13.26791)). SBC is a particular mechanism by which models could cooperate in equilibrium. Another aspect not mentioned in our original submission (but mentioned in footnote 1 of the Conitzer paper) is that cooperation between AIs can also be undesirable in some circumstances. For example, some AI alignment approaches, such as debate or automated interpretability, depend on checks and balances between different AI systems. Similarly, many real-world economic mechanisms such as auctions break if participants are able to collude. While most of our work is motivated by enabling SBC in contexts where cooperation is desirable, better understanding SBC and the conditions under which it can arise will also help us guard against such unintended collusion between models.

---

> > ### Comment · Reviewer_EVhh · 2023-08-14
> > **The devil is in the details...**
> >
> >
> > Thank you for the response.  When defining a diff function, I tend to think that "the devil is in the details."  Certainly in human interactions, agents who profess to follow a moral code, but then violate that code when they think they can get away with it, is a very common occurrence.  :-)
> >
> > However, basic theoretical work which shows that equilibria exist in simple settings is a first step towards solving that problem.
> >
> > Future work might also focus on how robust those equilibria are.  For example, if the diff function is not entirely reliable, i.e. reflecting a judgment that "the other agent looks honest, but I can't be sure", do the equilibria still exist?  How much noise or uncertainty can they tolerate before things fall apart?

---

> > > ### Author Response · Authors · 2023-08-21
> > > **(Optional) further thoughts on more realistic theoretical models and experiments**
> > >
> > > Thanks for this response! We agree with the reviewer’s comments. In particular, we agree that this is an important question and that future work should tackle it. We give some further thoughts on settings in which to investigate these things below. The reviewer and AC shouldn’t feel obligated to read these thoughts.
> > >
> > > It seems that an important question is whether _realistic_ ways of perceiving similarity are sufficient for establishing cooperative equilibrium. Both our theoretical results and our empirical results are about artificial ways of perceiving similarity. That being said, the reviewer's comment inspired us to consider the following slight extension of our model:
> > >
> > > Consider the Prisoner’s Dilemma, parameterized by G, as per Table 1 of our paper. Imagine that each player is only allowed to submit a threshold agent $(C,\theta_i,D)$. Now imagine that each player $i$’s diff function works as follows:
> > > With probability $p_i$: $\mathrm{diff}_i(\theta_1,\theta_2)=|\theta_1-\theta_2|+N$ with $N\sim \mathrm{Unif}([0,\epsilon])$ for some $\epsilon>0$.
> > > But with probability $1-p_i$: $\mathrm{diff}_i(\theta_1,\theta_2)=0$.
> > > (For $p_1,p_2=1$ this is exactly the setting in Example 1 and Proposition 1 of the paper.)
> > >
> > > Intuitively, this is supposed to model the case where each player can try to “manipulate” the diff value to be 0 and the manipulation succeeds with probability $1-p_i$. It is furthermore assumed (somewhat unrealistically) that if manipulation fails, the other player never learns of the attempt to manipulate. Instead, the diff value is observed normally if manipulation fails. That way we can assume that each player always attempts to manipulate.
> > >
> > > Then the following generalization of Proposition 1 holds:
> > > In the aforedescribed game, $((C,\theta_1,D),(C,\theta_2,D))$ is an equilibrium if and only if
> > > * $\theta_1,\theta_2 \leq 0$, or
> > > * $0\leq \theta_1=\theta_2 \leq \epsilon$ and $Gp_i\geq 2$ for $i=1,2$.
> > >
> > > The proof works just the same as the proof of the original Proposition 1. Intuitively, the idea (for the second part) is that if Player $i$ decreases $\theta_i$, their own probability of cooperation decreases by $2\delta/\epsilon$. Meanwhile, the opponent’s probability of cooperation decreases by $p_{-i} \delta/\epsilon$. Thus, the overall increase in Player $i$’s utility is $ 2\delta/\epsilon - G p_{-i} \delta/\epsilon$. For this to be nonnegative, we need to have $G*p_i\geq 2$.
> > >
> > > In terms of more realism, here is an ambitious setting that it would be exciting to study in future work. Say we have two complex base models. For concreteness, let’s say these are two potentially very different LLMs. Now, we get to train the LLMs to do SBC with each other. Call the resulting models the SBC base models. Now two players (principals) Alice and Bob play a game in which each chooses a fine-tuning scheme for one of the SBC base models. The LLMs then play a game against each other on the principals’ behalf. Let’s imagine that this game is a social dilemma. But let’s also imagine that this game has some player-specific aspects. In particular, imagine that Alice and Bob need to do some fine-tuning to achieve high utility and that this necessary fine-tuning is different between the two players. Finally, as our diff, we imagine that the two models make some observation about how they were fine-tuned. For example, we might imagine that they directly observe each other’s fine-tuning data sets. (At least for practical purposes this is still very far from full transparency.) But we might also imagine that they observe some abstract description of these data sets, such as a description of what kind of data they contain. For example, we might imagine descriptions such as, “The opponent model was trained on analyses of 18th century landscape paintings; it wasn’t trained on any strategic data” or “Alice’s and Bob’s fine-tuning data seems to have a very similar effect on strategic behavior”. We might use another language model to write a description of how the fine-tuning data sets relate.
> > >
> > > Now the challenge would be: Can we train the SBC base models in such a way that the game between Alice and Bob has cooperative (approximate) equilibria? We conjecture that the answer is yes. However, depending on various details of the setting, this is not at all obvious. For example, it may be possible for Alice or Bob to deploy “deceptive training data”, i.e., training data that looks innocuous but somehow still affects the strategic behavior.
> > >
> > > While this setting is still artificial (like most work on safety in social dilemmas between autonomous agents), we think it could elucidate the kind of worries shared by EVhh and the authors in a way that our numeric diff model cannot. Unfortunately, it is also vastly more complicated than training models with ~10k parameters and a handful of inputs and outputs. Nonetheless, we prospectively would like to investigate this kind of setting.

---

### Official Review · Reviewer_2ynN · 2023-07-06

**Soundness:** 3 good
**Presentation:** 4 excellent
**Contribution:** 2 fair
**Rating:** 6
**Confidence:** 3

**Summary:**

This paper considers the problem of two agents trained by machine learning who interact in a social dilemma.  The agents are able to observe a numeric measure of the similarity of their learned policies, and condition on this similarity when choosing their actions (in contrast to the "full transparency" case, in which the agents can each observe the other's full source code).  When policies can condition on similarity, cooperation can be supported in equilibrium (similarly to various full transparency results in the literature).

When partially-transparent policies are naively trained, they tend not to learn to play toward cooperative equilibria; however, the paper demonstrates that pre-training by averaging between playing against the same policy and a randomly-sampled policy can enable machine learned models to find partially cooperative equilibria.

**Strengths:**

This paper correctly observes that partial transparency supports the same equilibrium outcomes as full transparency, while being dramatically more plausible.  The results are carefully derived and clearly presented.  The paper studies a possible approach to solving an important problem; social dilemmas succinctly describe a fundamental problem in many situation where agents would all gain by cooperating.

This paper takes a step beyond proving the existence of equilibria, and demonstrates that plausible training procedures can actually find them.

The notion of similarity is based on the similarity of the policies considered as functions, rather than being computed on weights or some other parametric representation.  This is very natural, and I think it's an improvement over the "source code" based approach that earlier full-transparency literature tends to appeal to.

**Weaknesses:**

My main concern is that the results really only apply to a very specific class of games (player-symmetric additively decomposable games).  It makes sense that the equilibrium uniqueness results of theorem 4 should be narrowly targeted (it's hard to prove things about large classes of games), but I would have liked to see some experimental examination of whether the proposed approach works well on games that don't satisfy these restrictions.

The procedure described still tends not to find fully cooperative equilibria, which seems like a shortcoming.


#### Minor issues (did not affect rating)

- p.1: "While partial transparency is the norm": This is a pretty strong claim, and one which is important for the main claims of the paper.  Could you back it up with a citation or example or something else beyond a bare assertion?
- p.2: "In particular, full transparency can make the problem of equilibrium selection harder": Isn't this also true of partial transparency?



**Questions:**

Have you evaluated your approach on non-additively-decomposable games?

---

> ### Author Rebuttal · Authors · 2023-08-10
>
> We thank Reviewer 2ynN for their detailed review!
>
> >My main concern is that the results really only apply to a very specific class of games (player-symmetric additively decomposable games). It makes sense that the equilibrium uniqueness results of theorem 4 should be narrowly targeted (it's hard to prove things about large classes of games), but I would have liked to see some experimental examination of whether the proposed approach works well on games that don't satisfy these restrictions.
> >
> >Have you evaluated your approach on non-additively-decomposable games?
>
> Regarding further experiments on games that break our restrictions: We agree that it is important to study more types of games. We have run some informal experiments on games that are only approximately additively decomposable and only approximately symmetric (while using diff functions that ignore these slight asymmetries). Generally the results in these settings are – unsurprisingly – similar to the current setting. We are also currently working on a second project about similarity-based cooperation that studies more radically different games (such as Chicken, repeated games) and explores more different methods (mostly other methods from the opponent shaping family). We have decided to leave further experiments on other types of games to a different paper for the following reasons.
>
> Most importantly and mundanely, any new experiments would require extra space to describe, especially if new conceptual issues need to be addressed. For example, for an experiment with non-additively decomposable games, we would have to first define such a game a la HDPD (assuming we want the experiments to be comparable), which is non-trivial. We would also need to somehow address the difficulties of using CCDR in non-additively-decomposable games. (It may work very well without any specific modifications, e.g., in Chicken, but we would have to explain why it works.) Using the Prisoner’s Dilemma as a running example throughout the paper provides intuition for the more complicated HDPD. If we use a substantially new game – say, Chicken – we would have to first build up intuition for it. (What do we even expect the SBC equilibria to look like? Etc.) We don’t think we have the necessary space to spare to do all this – one could argue that we already rely too much on discussing details and results in the appendix that should be discussed in the main text (e.g., the LOLA results).
>
> We also think that many such further experiments are a more natural fit for future projects than they are for the present one. Generally, the main research question behind the experiments in the present paper is whether SBC equilibria in complex settings (as opposed to the very simple settings studied by theory) can be found with machine learning. It’s not clear a second example like a high-dimensional version of Chicken would add much to answering this question (at least if the complexities involved are very similar to those introduced by the HDPD). Secondarily, because the HDPD is closely related to our theoretical settings, our experimental results test whether our theoretical results apply in a more complex setting. It seems more natural to study other game-theoretic dynamics such as Chicken in contexts where one can ask more concrete questions. For example, if one could get opponent shaping to work in diff games, one could ask whether opponent shaping methods learn (non-diff-based) asymmetric Dare–Swerve equilibria or diff-based Swerve–Swerve equilibria.
>
> >p.1: "While partial transparency is the norm": This is a pretty strong claim, and one which is important for the main claims of the paper. Could you back it up with a citation or example or something else beyond a bare assertion?
>
> We will clarify the claim mentioned on p.1 – thanks for pointing it out! We here merely mean that it is common to have some non-trivial information about the other player other than that they are a rational agent with a particular utility function and particular beliefs. We do not yet want to make a claim at this point about how commonly this information induces new, better equilibria.
>
> >p.2: "In particular, full transparency can make the problem of equilibrium selection harder": Isn't this also true of partial transparency?
>
> Regarding the claim on p. 2: Yes, that’s correct and we’ll need to clarify this as well in the paper. What we meant to refer to here is that full transparency makes the equilibrium selection problem very difficult; diff games seem to avoid this under some assumptions. Our paper does not say anything about other kinds of partial transparency and what they do to the equilibrium selection problem. All of this is unclear at this point in the paper.

---

> > ### Comment · Reviewer_2ynN · 2023-08-14
> >
> > Thanks for the clarifications!

---

### Author Rebuttal · Authors · 2023-08-10

We thank the reviewers for their efforts in evaluating and helping us improve our paper! We are pleased that they found our paper “well written” (EVhh, ty7R), “easy to read and understand” (xnVF) and the results “clearly presented” (2ynN). It’s encouraging that the reviewers find our approach “interesting” (xnVF), “natural” (2ynN), “logical” (ty7R) as well as “more practical and relevant for AI” (ty7R) and “dramatically more plausible” (2ynN) than prior work on program equilibrium. We are also glad that the reviewers find our results “carefully derived” (2ynN), “solid” (EVhh) and “rigor[ous]” (xnVF), and that they appreciate our “excellent discussion of the limitations of CCDR” (EVhh) and the HPDP as “a cool independent contribution” (ty7R).

We will post replies to the individual reviews. We here address a question brought up by both xnVG and EVhh. Roughly, both xnVG and EVhh would have liked more discussion of realistic settings for SBC. They are also both concerned that the similarity metric could be gamed.

We agree that these questions are important. We believe that SBC has future applications to high-stakes interactions between AI systems, but recognize that there are many open questions about SBC. First, depending on how hard it is to find SBC equilibria, SBC may or may not play an important role. Our paper makes progress on this question and we hope future work will further resolve uncertainty. Second, we don’t know how future deployments of AI will go. For example, SBC is in part motivated by the belief that in future deployment scenarios, AI models will often strategically interact with near copies of each other (see more below). Perhaps future deployment will work differently. Due to limited space we have discussed these issues only briefly in our submission (e.g., lines 53–56), but we will elaborate here and in the camera ready copy as space permits.

First, we agree that gaming of the diff function is a concern. In fact, we think there are two somewhat different concerns.

1. As in program equilibrium, the policies in our setup are essentially commitment mechanisms that are made credible by the diff function. For example, let’s say that Alice submits a “Cooperate against copies; defect against non-copies” policy and diff reveals whether Alice’s and Bob’s policies are copies of each other. Then if Bob submits the same policy, he effectively credibly commits to that policy. Because we can view the policies and the diff observation as credible commitments, many of the usual points about commitment apply: Players would want to pretend to make credible commitments and try to renege on commitments if possible. For instance, in the above example, once the signal “Alice’s and Bob’s policies are copies” has been sent to Alice’s policy, Bob will want to intervene to defect, rather than follow his submitted policy. If this is possible, the cooperative equilibrium disappears.

2. Under a given diff function, it may be unclear whether a particular policy profile (\pi_1,\pi_2) is a cooperative equilibrium or not, even if the diff function is faithfully observed. For example, Player 1 might not know whether \pi_1 is exploitable.

We now argue that nonetheless SBC is applicable in the real world. (Some of the ideas below resemble ideas from discussions of other forms of credible commitment.) Imagine that A and B each delegate their choice in a complicated strategic scenario to an AI system. To do so, they both provide some information on the scenario they face and on their goals and the system implements or recommends some strategy to achieve these goals. Now imagine specifically that A and B both licensed their system from the same company and then fine-tuned it for their specific application area. Finally, imagine that the chatbots obtain this information (not necessarily from A and B). We think that SBC could apply in this type of scenario. That is, A’s and B’s systems can partially cooperate in this setting depending on how much domain-specific fine tuning occurs. For this we need some important but, we believe, often realistic assumptions:
- We suppose that A and B are unable to make a competent choice themselves (e.g., because the scenario is too complex, or because the AI system controls actuators that Alice and Bob cannot control themselves). So they cannot “intervene” to break SBC.
- We must also imagine that Alice and Bob cannot delegate to another AI system, at least without risking that the other learns of this switch. For example, we might imagine that they cannot afford to license multiple systems.
- We imagine that Alice and Bob cannot secretly modify their system to defect. For example, we may imagine that the AI company runs the system on their own servers and restricts the ways in which licensees can use or modify the system. Specifically, it may restrict deceptive applications including the ones needed to break SBC. Or we could again imagine that modifying the system’s strategic behavior would require a large effort that is hard to reliably keep secret.

One could also consider settings in which diff observation and other forms of partial transparency are set up deliberately, including to allow for cooperative equilibria. Cryptographic tools (computation on the blockchain, zero-knowledge proofs, etc.) provide interesting means to achieve this. More straightforwardly, a company or government entity could provide certifications of AI systems. For example, credibility in the above scenario could be achieved if the certification company ensures that A and B have not modified the strategic behavior of their AI systems w.r.t. SBC.

The above primarily addresses the first worry. They address the second worry insofar as it is intuitive that the examples of diff functions allow for equilibria. That said, it seems difficult to say for a given non-trivial, imprecise description of how similarity is observed whether the second type of gaming is possible or not.

---

### Decision · Program_Chairs · 2023-09-21

**Decision:**

Accept (poster)

**Comment:**

The paper has received a strong majority of supportive reviews. The authors are encouraged to address the comments of the reviewers as well as include the clarifications they provided during the discussion phase in improving the final version of their paper.